# COHERENCE-BASED LABEL PROPAGATION OVER TIME SERIES FOR ACCELERATED ACTIVE LEARNING

**Yooju Shin[1], Susik Yoon[2], Sundong Kim[3], Hwanjun Song[4], Jae-Gil Lee[1,*], Byung Suk Lee[5]**
[1]KAIST  [2]UIUC  [3]Institute for Basic Science  [4]NAVER AI Lab  [5]University of Vermont
{yooju.shin, jaegil}@kaist.ac.kr, susik@illinois.edu, sundong@ibs.re.kr,
hwanjun.song@navercorp.com, bslee@uvm.edu

## ABSTRACT

Time-series data are ubiquitous these days, but lack of the labels in time-series data is regarded as a hurdle for its broad applicability. Meanwhile, active learning has been successfully adopted to reduce the labeling efforts in various tasks. Thus, this paper addresses an important issue, *time-series active learning*. Inspired by the *temporal coherence* in time-series data, where consecutive data points tend to have the same label, our *label propagation* framework, called TCLP, automatically assigns a queried label to the data points within an accurately estimated time-series segment, thereby significantly boosting the impact of an individual query. Compared with traditional time-series active learning, TCLP is shown to improve the classification accuracy by up to 7.1 times when only $0.8\%$ of data points in the entire time series are queried for their labels.

## 1 INTRODUCTION

A *time series* is a sequence of data points at successive timestamps. Supervised learning (e.g., classification) with a time series requires the *label* of every data point, but unfortunately labels are often missing and hard to obtain due to lack of domain-specific knowledge (Shen et al., 2018; Malhotra et al., 2019; Li et al., 2020). It is worse for a time series collected for an extended length of time, as manually labeling so many data points is labor-intensive and time-consuming (Perslev et al., 2019; Tonekaboni et al., 2021). *Active learning* (Settles, 2009), a method that iteratively selects the most informative data point and queries a user for its label, can mitigate the high labeling cost. However, most active learning methods are *not* geared for time-series data, as they assume that data points are independent of one other (Sener & Savarese, 2018; Yoo & Kweon, 2019; Ash et al., 2020), which is obviously not true in time-series data.

Time-series data typically has the characteristic of *temporal coherence*; that is, temporally consecutive data points tend to have the same label (Wang et al., 2020; Barrow et al., 2020; Ishikawa et al., 2021). Let us refer to a sub-sequence of temporally coherent data points as a *segment*. For example, in motion-sensing time-series data, a segment consists of data points with the same motion status (e.g., walking, running). This temporal coherence of a segment can be exploited in time-series active learning. Specifically, when the label of a certain data point is obtained from a user, the same label can be propagated to other data points in the same segment. One challenge here is that the segment length is not known but needs to be estimated. If it is too short, unnecessarily frequent queries are issued; if too long, data points on the fringe of the segment are labeled incorrectly, consequently damaging the learning performance (e.g., classification accuracy). Thus, accurate estimation of the segments is important to enable the label propagation to achieve the maximum learning performance with the minimum number of queries.

This paper addresses the *label propagation segment estimation* problem in time-series active learning through a novel framework called *Temporal Coherence-based Label Propagation (TCLP)*. Figure 1 illustrates the overall workflow in the time-series active learning centered on TCLP. TCLP receives the class probabilities (i.e., softmax output) for the label of each data point from a classifier model and estimates the extent of the segment to propagate the label. This estimation is challenging

---

*Corresponding author.

Figure 1: Overall workflow of time-series active learning centered on TCLP. A data point that is queried for a label is determined by a query selection strategy. The label obtained is then propagated to adjacent data points guided by the TCLP framework.

in a real time series, as the classifier model output is uncertain and the time-series segments are unknown. TCLP meets this challenge by taking advantage of the temporal coherence via a quadratic plateau model (Moltisanti et al., 2019), by fitting it to the classifier model output to smooth out the fluctuations of class probabilities across consecutive data points.

To the best of our knowledge, TCLP is the first that performs label propagation for time-series active learning. The previous work closest to ours is pseudo-labeling in *single-timestamp supervised learning*, where labels are known for at least one data point in each segment (Moltisanti et al., 2019; Ma et al., 2020; Li et al., 2021). The approximate location and true class of a segment must be known in their work, which is often impractical in the real world. Moreover, the known labels are relatively dense in single-timestamp supervised learning, but they are very sparse in active learning—typically, no more than 5% of segments in our experiments. Thus, finding the boundaries between segments is more challenging in active learning than in single-timestamp supervised learning. To cope with the sparsity of labeled data points, TCLP performs *sparsity-aware* label propagation by exploiting temperature scaling (Guo et al., 2017) and plateau regularization.

Contributions of this paper are summarized as follows:

- It proposes a novel time-series active learning framework equipped with a sparsity-aware label propagation within an accurately estimated segment.
- It verifies the merit of TCLP through extensive experiments. The classification accuracy is improved by up to 7.1 times with TCLP compared to without label propagation. Moreover, TCLP works with any query selection strategy including core-set sampling (Sener & Savarese, 2018) and BADGE (Ash et al., 2020), boosting the effect of individual labeling.

## 2 RELATED WORK

### 2.1 ACTIVE LEARNING

Active learning is a special case of machine learning that 'actively' queries a user for the labels of data points, to the effect of using fewer labels to achieve the same learning performance. Recent studies have focused on developing such query strategies for machine learning based on deep neural networks (Settles, 2009; Ren et al., 2020). These approaches exploit prediction probabilities (Beluch et al., 2018), embeddings (Sener & Savarese, 2018), gradients (Ash et al., 2020), and losses (Yoo & Kweon, 2019) from deep neural networks to estimate the impact of each unlabeled data point if it were to be labeled. However, these methods are not suitable for time-series data, because they assume that data points are independent.

Several methods have been developed for time-series or sequence data, but most of them are applicable to only *segmented* time-series data under the assumption that a time series is already divided into labeled and unlabeled segments. Treating these segments as independent and identically distributed, these methods simply apply existing active learning frameworks to the segments. For example, He et al. (2015) select unlabeled segments that are far from labeled segments to maximize diversity; Peng et al. (2017) select unlabeled segments with distinctive patterns to maximize diversity; and Zhang et al. (2017) select unlabeled segments with high gradients to consider uncertainty for sentence classification. In addition, new neural network architectures or measures have been developed for sequence-data applications such as named entity recognition (Shen et al., 2018), video action recognition (Wang et al., 2018), and speech recognition (Malhotra et al., 2019). None of these methods is applicable to our problem, which handles *unsegmented* time-series data.

### 2.2 PSEUDO-LABELING

Pseudo-labeling has been actively studied for label-deficient learning environments, such as semi-supervised learning, to exploit unlabeled data points in training a classifier (Lee et al., 2013). In

general, a *pseudo-label* is given to an unlabeled data point based on the predictions from a classifier trained with labeled data points. Confidence-based methods create a pseudo-label if it is confidently predicted by a classifier (Lee et al., 2013). Consistency-based methods create a pseudo-label if it is consistently predicted for the original and augmented data points (Sajjadi et al., 2016; Rizve et al., 2021). Graph-based methods propagate pseudo-labels from labeled data points (nodes) to unlabeled data points based on a similarity graph constructed from the features of all data points (Shi et al., 2018; Liu et al., 2019; Wagner et al., 2018). However, these methods are not designed for time-series data, and therefore are not directly applicable to our problem.

Coherence-based methods are developed for *single-timestamp supervised learning* for unsegmented time-series data; they assume that at least one data point in each segment is given a true class label through weak-supervision. Ma et al. (2020) propose *probability thresholding propagation (PTP)*, which propagates known labels bidirectionally unless the predicted class probability for each data point is decreased by more than a threshold. Deldari et al. (2021) propose *embedding similarity propagation (ESP)*, which propagates known labels bidirectionally unless the embedding of each data point changes rapidly. Recently, Moltisanti et al. (2019) adopt a plateau model that represents class probabilities across consecutive data points, where a plateau model is constructed for each labeled data point and fitted to the classifier output; a known label is propagated as long as the value of a plateau model is higher than a threshold. While this work shares the idea of using a plateau model with our work, using the plateau model as it is for *active learning* results in performance degradation owing to the difference in the density of known labels, as will be shown in Section 4.

## 3 TCLP: TEMPORAL COHERENCE-BASED LABEL PROPAGATION

### 3.1 PRELIMINARIES AND PROBLEM SETTING

**Active learning:** Let $\mathcal{D} = \{(\boldsymbol{x}_t, y_t), t \in \mathcal{T}\}$ be a time series where $\mathcal{T}$ is the index set of timestamps; $\boldsymbol{x}_t$ is a multi-dimensional data point at timestamp $t$, and $y_t$ is one of the class labels if $\boldsymbol{x}_t$ is labeled or null otherwise. Let $\mathcal{D}_L \subseteq \mathcal{D}$ be a labeled set, i.e., a set of labeled data points, and $\mathcal{D}_U \subseteq \mathcal{D}$ be an unlabeled set, i.e., a set of unlabeled data points, where $D_U \cup D_L = D$. At each round of active learning, $b$ data points are selected from $\mathcal{D}_U$ by a query selection strategy, such as entropy sampling (Wang & Shang, 2014), core-set selection (Sener & Savarese, 2018), and BADGE (Ash et al., 2020), and their ground-truth labels are obtained from a user; these newly-labeled $b$ data points are then removed from $\mathcal{D}_U$ and added to $\mathcal{D}_L$. After $\mathcal{D}_L$ is updated, a classifier model $f_\theta$ is re-trained using the updated labeled set.

**Label propagation:** Given a data point at timestamp $t_q$ and its label, $(\boldsymbol{x}_{t_q}, y_{t_q})$, obtained from a user in response to a query, TCLP assigns the label $y_{t_q}$ to nearby data points in the timestamp range $[t_s : t_e]$ $(t_s \leq t_q \leq t_e)$ estimated according to its temporal coherence property criteria. We call the sub-sequence of data points in $[t_s : t_e]$ an *estimated segment* at $t_q$. There are two properties: (i) *accuracy*, which indicates that as many data points in the segment as possible should have the same ground-truth label $y_{t_q}$; and (ii) *coverage*, which indicates that the length of the segment $(t_e - t_s)$ should be as long as possible. More formally, we estimate the segment for $t_q$ by

$$t_s, t_e = \underset{t_s', t_e'}{\arg\min} \frac{1}{t_e' - t_s'} \sum_{t=t_s'}^{t_e'} \left(1 - f_\theta(\boldsymbol{x}_t)[y_{t_q}]\right), \tag{1}$$

where $t_s' \leq t_q \leq t_e'$ holds, $f_\theta(\boldsymbol{x}_t)$ is the softmax output vector of the classifier model at timestamp $t$, and $f_\theta(\boldsymbol{x}_t)[y_{t_q}]$ is the estimated probability of the label $y_{t_q}$. In Equation (1), the accuracy is achieved by minimizing the sum of errors in the numerator, and the coverage is achieved by maximizing the candidate segment length in the denominator. Note that estimated probabilities are used to calculate the errors, since the true probabilities are not known.

Once segment estimation is done, all data points in the estimated segment (i.e., in $[t_s : t_e]$) are removed from $\mathcal{D}_U$ and added to $\mathcal{D}_L$ with the label $y_{t_q}$, thus doing *coherence-based label propagation*. At each round of active learning, the *segment estimation* repeats for each of the $b$ queried data points; as a result, the size of $\mathcal{D}_L$ is increased by the total length of the estimated segments. Besides, we allow the data points in $[t_s : t_e]$, except $t_q$, to be queried again in subsequent rounds so that the propagated labels can be refined subsequently.

### 3.2 PLATEAU MODEL FOR SEGMENT ESTIMATION

An adequately trained classifier model $f_\theta$ returns higher probabilities of the (unknown) true labels for data points inside a segment and lower priorities for data points outside the segment. Besides, the output probabilities of the model are not constant within the segment because of noise in the time-series data. Thus, one natural approach to finding a segment is to fit a plateau model to the output of the classifier model and make a plateau of probability 1 into an estimated segment.

#### 3.2.1 PLATEAU MODEL AND ITS FITTING

Among many functions with a plateau-shaped value, we use the function introduced by Moltisanti et al. (2019):

$$h(t; c, w, s) = \frac{1}{(e^{s(t-c-w)} + 1)(e^{s(-t+c-w)} + 1)}, \quad (2)$$

where $c$, $w$, and $s$ are trainable parameters for the model $h$. As shown in Figure 2, $c$ and $w$ respectively represent the center and half-width of the plateau, and $s$ indicates the steepness of the side slopes.

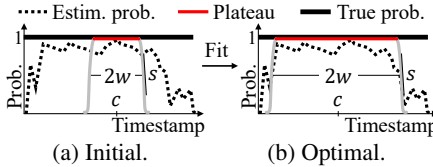

Figure 2: Plateau model and its fitting.

The fitting of the plateau model at timestamp $t_q$ is illustrated in Figure 2. At the beginning, as in Figure 2a, $c$ is set to $t_q$, and $w$ and $s$ are set to initial values and updated by the following optimization

$$c, w, s = \underset{c', w', s'}{\arg\min} \frac{1}{2w'} \sum_{t=c'-w'}^{c'+w'} |h(t; c', w', s') - f_\theta(\boldsymbol{x}_t)[y_{t_q}]|, \quad (3)$$

where $h(t; c', w', s') = 1$ for $t \in [c' - w' : c' + w']$. Letting $\epsilon(t)$ be $|h(t; c', w', s') - f_\theta(\boldsymbol{x}_t)[y_{t_q}]|/2w'$ in Equation (3) and $E$ be $\sum_t \epsilon(t)$, the optimal values of the three parameters are obtained by repeating a gradient update step,

$$c = c' - \eta \nabla_{c'} E, w = w' - \eta \nabla_{w'} E, \text{ and } s = s' - \eta \nabla_{s'} E, \text{ where } \eta \text{ is the learning rate.} \quad (4)$$

After a model is fitted through enough rounds, as shown in Figure 2b, the plateau is located at the center of a true segment and its width covers most of the true segment. $[c - w : c + w]$, indicated by the red line in Figure 2b, is determined as the estimated segment for the plateau model at $t_q$. Overall, as shown in Figures 3a and 3b, as active learning progresses, more data points are queried, estimated probabilities becomes more accurate, and plateau models are better fitted to the estimated probabilities. Eventually, the plateau models accurately represent the true segments in the time series.

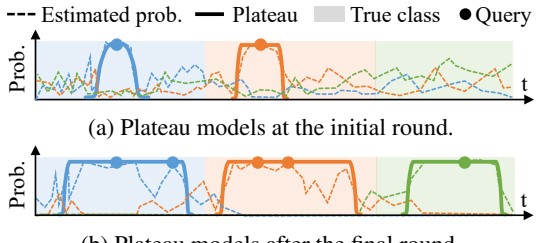

(a) Plateau models at the initial round.

(b) Plateau models after the final round.

Figure 3: Updated plateau models.

#### 3.2.2 SPARSITY-AWARE LABEL PROPAGATION

In active learning, known labels are typically very sparse—initially no more than 5% of the segments and growing slowly as more labels are obtained for queried data points. Our experience indicates that simply optimizing the plateau model as explained in Section 3.2.1 tends to generate plateaus much longer than true segments. See Section 4.4 for the poor performance of the typical (sparsity-unaware) plateau model. On the other hand, in single-timestamp supervised learning, where the plateau model has been successfully employed (Moltisanti et al., 2019), at least one label should exist for every segment; thus, the boundaries of segments can be easily recognized while making the plateaus not overlap between segments. Thus, to overcome the lack of potential boundaries for segment estimation, we extend the procedure of the plateau model fitting in three ways.

**Calibrating the classifier output:** Because modern neural network models have millions of learning parameters, the distribution of predicted probabilities in such models is often highly skewed to either 1 or 0, which means that the model is overconfident (Guo et al., 2017; Müller et al., 2019). That is, the output of the classifier model $f_\theta$ could be too high even outside the true segment, thereby making the

plateau wider than the true segment. We address this issue by employing *temperature scaling* (Guo et al., 2017) to calibrate the classifier output (i.e., softmax probabilities). Temperature scaling is widely used to mitigate the overconfidence issue, because it reduces the differences of the softmax probabilities while keeping their order. Specifically, temperature scaling divides the logits (inputs to the softmax layer) $\boldsymbol{z}_t$ by a learned scale parameter $T$, i.e., as $f_\theta(\boldsymbol{x}_t) = \text{softmax}(\boldsymbol{z}_t/T)$.

**Regularizing the width of a plateau:** To prevent a plateau from rapidly growing wider than a true segment, we constrain the amount of updates on the parameter $w$ of the plateau model. Specifically, $w$ cannot be increased more than twice its current value in a single round of active learning. Accordingly, the gradient update of $w$ in Equation (4) is slightly modified to $w = \min(2w', w' - \eta \nabla_{w'} \epsilon(t))$.

**Balancing the class skewness:** This issue of class label imbalance can become more severe by label propagation from sporadic queries; although the propagated labels are correct, the number of such propagated labels can vary across different classes. To reduce the effect of this potential skewness, we *re-weight* the loss $\ell(\hat{y}_t, y_t)$ at each timestamp in training the classifier model $f_\theta$, where $\hat{y}_t = \arg\max_k f_\theta(\boldsymbol{x}_t)[k]$ and $y_t$ is the propagated ground-truth label (Johnson & Khoshgoftaar, 2019). The loss is adjusted by the inverse ratio of the number of the timestamps of a given class over that of the most infrequent class. That is, if we let $N_k$ be the number of the timestamps assigned with the class $k$, the parameter of the classifier model is updated as follows: $\theta = \theta - \lambda \frac{\min_i N_i}{N_{y_t}} \nabla \ell(\hat{y}_t, y_t)$, where $\lambda$ is another learning rate for training a classifier model $f_\theta$.

## 3.3 THEORETICAL ANALYSIS

We show that our plateau-based segment estimation is expected to produce a segment closer to the true segment than a simple threshold-based segment estimation. For ease of analysis, we consider a single segment whose true length is $L$, the query data point at $t_q$ is located at the center of the true segment with $k = y_{t_q}$ known. In addition, we assume that the estimated class probabilities $f_\theta(\boldsymbol{x}_t)[k](1 \leq t \leq L)$ are conditionally independent at different timestamps (Graves et al., 2006; Chung et al., 2015).

**Threshold-based segment estimation:** A simple and straightforward way to estimate the segment is to expand its width bidirectionally as long as the estimated probability at each timestamp is higher than or equal to a threshold $\delta$. The probability that the length of a segment reaches $l$ is

$$\Pr(t_e - t_s = l) = l \cdot z^{l-1}(1-z)^2, \tag{5}$$

where $z = \Pr(f_\theta(\boldsymbol{x}_t)[k] \geq \delta)$. Here, the $l$ multiplied to $z^{l-1}(1-z)^2$ in Equation (5) is the number of alignments possible for a segment containing the center $t_q$. As a result, the expected length is

$$\mathbb{E}_{f_\theta(\boldsymbol{x}_t)}[t_e - t_s] = \sum_{l=1}^{L} l \cdot \Pr(t_e - t_s = l) = \sum_{l=1}^{L} l^2 z^{l-1}(1-z)^2$$
$$= \frac{1 + z - (L+1)^2 z^L + (2L^2 + 2L - 1)z^{L+1} - L^2 z^{L+2}}{1-z}. \tag{6}$$

**Plateau-based segment estimation:** Let us fix $c$ to $t_q$ and $s$ to $\infty$ (90° steepness), and denote the plateau model simply as $h(t; w)$. Then, $h(t; w) = 1$ if $c - w \leq t \leq c + w$, and $h(t; w) = 0$ otherwise. In addition, for simplicity, let us fix the denominator $2w'$ in Equation (3) to $L$. Then, the inside of the argmin operator in Equation (3) becomes

$$\epsilon(l) = \frac{1}{L}\big((L - l) \cdot |0 - f_\theta(\boldsymbol{x}_t)[k]| + l \cdot |1 - f_\theta(\boldsymbol{x}_t)[k]|\big), \tag{7}$$

where $l(= 2w)$ denotes the length of the segment estimated with the plateau model. $\epsilon(l)$ is a linear function of $l$, where its slope is $(1 - 2f_\theta(\boldsymbol{x}_t)[k])$ and $1 \leq l \leq L$. Thus, Equation (7) evaluates to the minimum at either $l = 1$ when the slope is positive or $l = L$ when the slope is negative. Letting $z$ be $\Pr(f_\theta(\boldsymbol{x}_t)[k] \geq 0.5)$, the probabilities of $l = 1$ and $l = L$ are $1 - z$ and $z$, respectively. In conclusion, the expected length of the estimated segment is

$$\mathbb{E}_{f_\theta(\boldsymbol{x}_t)}[t_e - t_s] = 1 \cdot \Pr(l = 1) + L \cdot \Pr(l = L) = 1 - z + Lz. \tag{8}$$

**Comparison and discussion:** As $L$ increases, Equation (6) converges toward $\frac{1+z}{1-z}$ and is not affected by $L$, whereas Equation (8), which equals $z(L-1) + 1$, increases linearly with $L$. Therefore, when

a true segment is sufficiently long and $z$ is in the typical range (e.g., less than 0.9), the plateau-based segmentation (Equation (8)) is expected to produce a longer (i.e., closer to $L$) segment than the threshold-based segmentation (Equation (6)).

## 3.4 OVERALL ACTIVE LEARNING PROCEDURE WITH TCLP

---

**Algorithm 1** Time-series active learning with TCLP

---

**Input:** Timestamp feature $\boldsymbol{x}_t$, initially labeled set $\mathcal{D}_L$, unlabeled set $\mathcal{D}_U$, query strategy $\mathcal{Q}$,
    number of rounds $R$, query size $b$, initial classifier $f_{\theta_0}$, classifier loss $\ell$, learning rate $\lambda$.

**Output:** Final classifier $f_{\theta_{R+1}}$.

  1: $\mathbb{H}_0 \leftarrow$ Initialized plateau models for $\mathcal{D}_L$;

  2: $\theta_1 = \theta_0 - \lambda \frac{\min_i N_i}{N_{y_t}} \nabla \ell(\hat{y}_t, y_t)$ for each $(\boldsymbol{x}_t, y_t) \in \mathcal{D}_L$;

  3: **for** $r = 1, \ldots, R$ **do**

  4:     $\bar{\boldsymbol{y}}_t = \text{TEMPERATURESCALING}(f_{\theta_r}(\boldsymbol{x}_t))$; $\mathbb{H}_r = \varnothing$;        // see Section 3.2.2

  5:     **for** $h$ in $\mathbb{H}_{r-1}$ **do**

  6:         $h' \leftarrow$ fit $h$ on $\bar{\boldsymbol{y}}_t$; $\mathbb{H}_r \leftarrow \mathbb{H}_r \bigcup h'$;        // see Section 3.2.1

  7:     $\{t_q\}_{q=1}^b \leftarrow$ queried timestamps acquired by $\mathcal{Q}$ from $\mathcal{D}_U$;

  8:     $\mathbb{H}_r \leftarrow \mathbb{H}_r \bigcup \{\text{plateau model } h_{t_q}\}_{q=1}^b$;

  9:     $\mathbb{H}_r \leftarrow \text{ADJUST}(\mathbb{H}_r)$;        // see Appendix C

10:     $\mathcal{D}_L \leftarrow \text{LABELPROPAGATION}(\mathbb{H}_r)$;        // see Section 3.2.2

11:     $\theta_{r+1} = \theta_r - \lambda \frac{\min_i N_i}{N_{y_t}} \nabla \ell(\hat{y}_t, y_t)$ for each $(\boldsymbol{x}_t, y_t) \in \mathcal{D}_L$;

12: **return** $f_{\theta_{R+1}}$;

---

Algorithm 1 summarizes how TCLP works in time-series active learning. First, plateau models are initialized with the initially labeled set $\mathcal{D}_L$ and stored in the set $\mathbb{H}_0$ (Line 1); and then using $\mathcal{D}_L$, TCLP trains the classifier model $f_{\theta_0}$ (Line 2). Then, at each active learning round $r$, TCLP first performs calibration by inferring the data points with the classifier model $f_{\theta_r}$ and scaling the softmax output, and then initializes a new set of plateau models $\mathbb{H}_r$ (Line 4). Next, each plateau model in $\mathbb{H}_{r-1}$ from the previous round is fitted to the scaled output, and then the updated plateau model is added to $\mathbb{H}_r$ (Line 6). The new plateau models are then initialized from queried timestamp labels (Line 7) and added to $\mathbb{H}_r$ (Line 8). Then, any overlapping plateaus in $\mathbb{H}_r$ are adjusted—either merged into one or reduced to avoid the overlap—as needed (Line 9). Finally, the queried labels are propagated following the plateau models in $\mathbb{H}_r$ (Line 10), and the classifier model $f_{\theta_r}$ is re-trained with the augmented labeled set $\mathcal{D}_L$ (Line 11). The complexity analysis of TCLP is presented in Appendix A.

# 4 EVALUATION

We conduct experiments with various active learning settings to test the following hypotheses.

- TCLP accelerates active learning methods faster than other label propagation methods can.
- TCLP achieves both high accuracy and wide coverage in segment estimation.
- TCLP overcomes the label sparsity by the extensions discussed in Section 3.2.2.

## 4.1 EXPERIMENT SETTING

**Datasets:** The four benchmark datasets summarized in Table 1 are used. 50Salads contains videos at 30 frames per second that capture 25 people preparing a salad (Stein & McKenna, 2013), and GTEA contains 15 frame videos of four people (Fathi et al., 2011). For these two video datasets, we extract I3D features of 2,048 dimensional vectors at each timestamp follow-

Table 1: Summary of datasets and configurations.

| | Timestamps | Length | #class | Dim | $b$ | $R$ | $w_0$ | $s_0$ |
|---|---|---|---|---|---|---|---|---|
| 50salads | 288798 | 289 | 19 | 2048 | 200 | 15 | 15 | 0.5 |
| GTEA | 31225 | 34 | 11 | 2048 | 200 | 15 | 5 | 0.5 |
| mHealth | 343195 | 2933 | 12 | 23 | 200 | 15 | 15 | 0.5 |
| HAPT | 815614 | 967 | 6 | 6 | 200 | 15 | 15 | 0.5 |

ing the previous literature (Farha & Gall, 2019). mHealth contains 50Hz sensor time-series recordings of human movement, measured by 3D accelerometers, 3D gyroscopes, 3D magnetometers, and electrocardiograms (Banos et al., 2014); we extract labeled regions from the raw data and stitch

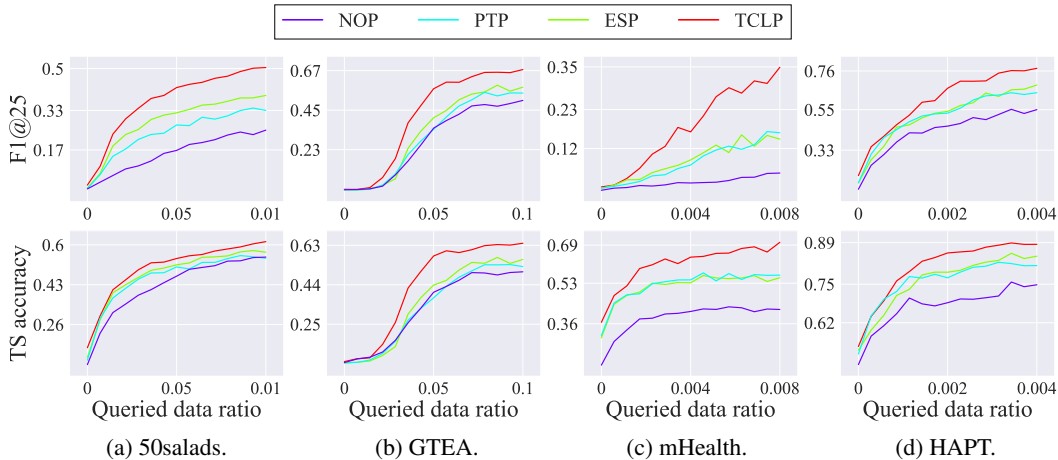

Figure 4: Classification accuracy measured at each (1st–15th) round of active learning. The accuracy value is an average over all query selection methods. Detailed results are in Appendix B.

them in chronological order to make a time series. HAPT represents 50Hz sensor time-series recordings of human actions in laboratory setting, measured by 3D accelerometers and multiple 3D gyroscopes (Anguita et al., 2013).

**Query selection methods:** To evaluate the efficacy of TCLP, we combine it with six different query selection methods. CONF (Wang & Shang, 2014) selects $b$ timestamps exhibiting the lowest confidence in the model's prediction, where the confidence is evaluated by using the largest predicted class probability; MARG (Settles, 2012) is similar to CONF, but it defines the confidence as the difference between the first- and second-largest predicted class probabilities; ENTROPY (Wang & Shang, 2014) selects the top $b$ timestamps exhibiting the largest entropy for their predicted class probabilities; CS (Sener & Savarese, 2018) chooses the top $b$ most representative timestamps in embedding space; BADGE (Ash et al., 2020) computes gradients from $f_\theta$ at each timestamp $t$ and queries $b$ timestamps found by $k$-MEANS++ to consider uncertainty and diversity; UTILITY is our simple selection strategy that selects $b$ timestamps randomly from the timestamps not covered by the current set of plateau models, to increase the utility of the plateau models.

**Compared label propagation methods:** For a thorough comparison, we compare TCLP with three available label propagation approaches—NOP, PTP, and ESP. For this purpose, each of TCLP and the three approaches is combined with each of the aforementioned six query selection methods. NOP is the baseline without using any label propagation. As explained in Section 2.2, PTP propagates labels based on the predicted class probabilities with a certain threshold ($\delta = 0.8$), while ESP leverages cosine similarity between embeddings for label propagation. PTP and ESP are modified to work in an active learning setting as done by Deldari et al. (2021).

**TCLP implementation details:** We use the multi-stage temporal convolutional network (MS-TCN) (Farha & Gall, 2019) as the classifier $f_\theta$ for time-series data. We use exactly the same training configuration suggested in the original work (Farha & Gall, 2019). Regarding active learning hyperparameters, the number of queried data points per round ($b$) and the number of active learning rounds ($R$) are summarized in Table 1. For TCLP, we use the initial parameters for plateau models ($w_0$ and $s_0$) in Table 1 and temperature scaling with $T = 2$. Our experience indicates that any value of $w_0$ smaller than 20% of the mean segment length is adequate enough. **Accuracy metrics:** *Timestamp accuracy* and *segmental F1 score* are measured at each round by five-fold cross validation; they represent the prediction accuracy at the granularity of *timestamp* and *segment*, respectively. The former is defined as the proportion of the timestamps with correct prediction. The latter is defined as the F1 score of segments with an overlapping threshold on the intersection over union (IoU) (Farha & Gall, 2019); that is, a prediction is classified as correct if the IoU between predicted and true segments is larger than the threshold. F1@25, with the threshold 25%, is commonly used in the literature (Lea et al., 2017; Farha & Gall, 2019); the trends with the thresholds 10% and 50% are similar.

Table 2: Classification accuracy measured after the final (15th) round (the best results in bold).

| Dataset | Query | F1@25 | | | | Timestamp Accuracy | | | |
|---|---|---|---|---|---|---|---|---|---|
| | | NOP | PTP | ESP | **TCLP** | NOP | PTP | ESP | **TCLP** |
| 50salads | CONF | 0.191±0.015 | 0.204±0.015 | 0.280±0.017 | **0.433±0.010** | 0.505±0.017 | 0.451±0.032 | 0.462±0.031 | **0.559±0.010** |
| | ENTROPY | 0.133±0.004 | 0.193±0.011 | 0.263±0.020 | **0.368±0.031** | 0.432±0.019 | 0.416±0.019 | 0.455±0.024 | **0.496±0.027** |
| | MARG | 0.287±0.021 | 0.359±0.018 | 0.436±0.031 | **0.600±0.028** | 0.616±0.033 | 0.615±0.015 | 0.637±0.031 | **0.697±0.020** |
| | CS | 0.322±0.021 | 0.426±0.018 | 0.480±0.022 | **0.559±0.021** | 0.595±0.021 | 0.602±0.017 | 0.632±0.023 | **0.657±0.024** |
| | BADGE | 0.197±0.014 | 0.317±0.032 | 0.377±0.019 | **0.471±0.030** | 0.514±0.023 | 0.529±0.024 | 0.567±0.018 | **0.600±0.025** |
| | UTILITY | 0.372±0.017 | 0.482±0.012 | 0.511±0.028 | **0.595±0.022** | 0.625±0.028 | 0.642±0.022 | 0.659±0.026 | **0.672±0.018** |
| | AVERAGE | 0.250±0.034 | 0.330±0.043 | 0.391±0.038 | **0.504±0.035** | 0.548±0.028 | 0.543±0.035 | 0.569±0.034 | **0.614±0.028** |
| GTEA | CONF | 0.386±0.119 | 0.297±0.123 | 0.443±0.108 | **0.690±0.020** | 0.409±0.107 | 0.321±0.115 | 0.443±0.082 | **0.654±0.011** |
| | ENTROPY | 0.355±0.119 | 0.575±0.032 | 0.422±0.128 | **0.613±0.028** | 0.364±0.114 | 0.565±0.028 | 0.456±0.104 | **0.590±0.021** |
| | MARG | 0.248±0.072 | 0.491±0.122 | 0.582±0.035 | **0.727±0.024** | 0.320±0.057 | 0.469±0.099 | 0.545±0.026 | **0.659±0.015** |
| | CS | 0.656±0.031 | 0.512±0.125 | 0.610±0.057 | **0.666±0.011** | 0.609±0.026 | 0.520±0.102 | 0.591±0.035 | **0.630±0.007** |
| | BADGE | 0.718±0.041 | **0.723±0.033** | 0.710±0.029 | 0.703±0.028 | **0.705±0.020** | 0.682±0.010 | 0.692±0.017 | 0.663±0.015 |
| | UTILITY | 0.661±0.025 | 0.671±0.030 | **0.700±0.014** | 0.656±0.025 | 0.608±0.018 | 0.608±0.033 | **0.646±0.019** | 0.644±0.019 |
| | AVERAGE | 0.504±0.074 | 0.545±0.056 | 0.578±0.046 | **0.676±0.015** | 0.502±0.059 | 0.528±0.047 | 0.562±0.038 | **0.640±0.010** |
| mHealth | CONF | 0.016±0.005 | 0.015±0.002 | 0.029±0.006 | **0.228±0.064** | 0.294±0.016 | 0.445±0.044 | 0.398±0.063 | **0.685±0.051** |
| | ENTROPY | 0.008±0.004 | 0.011±0.001 | 0.023±0.005 | **0.074±0.030** | 0.175±0.047 | 0.363±0.032 | 0.413±0.050 | **0.560±0.054** |
| | MARG | 0.018±0.005 | 0.065±0.017 | 0.078±0.027 | **0.363±0.082** | 0.305±0.011 | 0.485±0.042 | 0.485±0.030 | **0.693±0.079** |
| | CS | 0.055±0.024 | 0.289±0.085 | 0.187±0.049 | **0.594±0.094** | 0.590±0.034 | 0.680±0.032 | 0.577±0.040 | **0.817±0.009** |
| | BADGE | 0.131±0.042 | 0.145±0.041 | 0.129±0.024 | **0.296±0.054** | 0.404±0.056 | 0.578±0.026 | 0.545±0.049 | **0.665±0.047** |
| | UTILITY | 0.076±0.006 | 0.462±0.103 | 0.432±0.089 | **0.538±0.113** | 0.752±0.051 | 0.831±0.028 | **0.901±0.015** | 0.789±0.065 |
| | AVERAGE | 0.051±0.018 | 0.164±0.067 | 0.146±0.057 | **0.349±0.072** | 0.420±0.080 | 0.564±0.064 | 0.553±0.069 | **0.702±0.034** |
| HAPT | CONF | 0.324±0.114 | 0.289±0.084 | 0.459±0.050 | **0.656±0.034** | 0.611±0.107 | 0.592±0.083 | 0.793±0.026 | **0.857±0.019** |
| | ENTROPY | 0.332±0.055 | 0.368±0.082 | 0.384±0.044 | **0.653±0.047** | 0.646±0.067 | 0.755±0.024 | 0.763±0.044 | **0.836±0.011** |
| | MARG | 0.276±0.043 | 0.667±0.042 | 0.790±0.020 | **0.805±0.022** | 0.556±0.031 | 0.799±0.010 | 0.809±0.025 | **0.889±0.031** |
| | CS | 0.739±0.033 | 0.816±0.010 | **0.843±0.026** | 0.835±0.024 | 0.880±0.005 | 0.898±0.018 | **0.926±0.007** | 0.909±0.014 |
| | BADGE | 0.784±0.029 | 0.719±0.044 | 0.813±0.019 | **0.860±0.018** | 0.859±0.032 | 0.840±0.016 | 0.858±0.014 | **0.886±0.035** |
| | UTILITY | 0.846±0.012 | **0.867±0.018** | 0.811±0.026 | 0.849±0.023 | 0.936±0.006 | **0.937±0.008** | 0.916±0.010 | 0.934±0.004 |
| | AVERAGE | 0.550±0.099 | 0.621±0.089 | 0.684±0.076 | **0.776±0.036** | 0.748±0.060 | 0.803±0.046 | 0.844±0.025 | **0.885±0.013** |

## 4.2 Overall Active Learning Performance

Figure 4 shows the F1@25 and timestamp accuracy at each round of varying the queried data ratio (= the number of queried data points / the total number of data points), where the accuracy values are averaged over the six query selection methods. 15 rounds are conducted for each dataset. TCLP performs the best among all label propagation approaches, with the accuracy improving much faster with a smaller number of queries than in the other approaches; this performance is attributed to the larger number of correctly propagated labels in TCLP, as will be shown in Section 4.3. Interestingly, the accuracy gain is higher in F1@25 than in timestamp accuracy; this difference makes sense because F1@25 measures the accuracy at the granularity of segment and therefore reflects temporal coherence better than the granularity of timestamp. Appendix D shows more details.

Table 2 shows the F1@25 and timestamp accuracy measured after the final (15th) round, i.e., at the last queried data ratio in Figure 4, for each of the six query selection methods. TCLP performs best here as well in almost all combinations of datasets, query selection methods, and accuracy metrics. Specifically, TCLP outperforms the compared label propagation approaches (NOP, PTP, and ESP) for all query selection methods except only a few cases. This result confirms that TCLP maintains its performance advantage regardless of the query selection method. Interestingly, TCLP's performance gain is most outstanding for the mHealth dataset and least outstanding for the GTEA dataset. The reason lies in the length of the segments. As shown in Table 1, mHealth's segments are the longest (2,933 on average) and GTEA's segments are the shortest (34 on average). Longer segments certainly allow more temporal coherence to be exploited in label propagation, thus resulting in higher performance. For instance, using UTILITY on the mHealth dataset, TCLP outperforms NOP by 7.1 times, PTP by 1.2 times, and ESP by 1.2 times in F1@25 while, on the GTEA dataset, TCLP outperforms them less significantly.

## 4.3 Label Propagation Performance

Table 3 shows the correct label propagation ratio (= the number of correctly propagated labels / the total number of data points) to verify how many labels are correctly propagated with each label propagation approach. Overall, fully taking advantage of the temporal coherence based on the plateau model, TCLP adds far more correct labels than PTP and ESP. Specifically, using UTILITY on the mHealth dataset, the correct propagation ratio of TCLP is higher than that of PTP by 5.0 times and that of ESP by 3.7 times. It is impressive that querying only 0.8% of data points results in up to 33% of data points correctly labeled. Appendix E shows more details.

Table 3: Correct label propagation ratio after the final (15th) round (the best results in bold).

| Dataset | Query | Correct Propagation Ratio | | | Dataset | Query | Correct Propagation Ratio | | |
|---|---|---|---|---|---|---|---|---|---|
| | | PTP | ESP | **TCLP** | | | PTP | ESP | **TCLP** |
| 50salads | CONF | 0.032±0.001 | 0.054±0.001 | **0.129±0.005** | mHealth | CONF | 0.027±0.001 | 0.038±0.002 | **0.109±0.017** |
| | ENTROPY | 0.026±0.000 | 0.042±0.001 | **0.100±0.005** | | ENTROPY | 0.024±0.000 | 0.031±0.001 | **0.066±0.009** |
| | MARG | 0.071±0.000 | 0.151±0.004 | **0.368±0.006** | | MARG | 0.054±0.001 | 0.083±0.004 | **0.204±0.027** |
| | CS | 0.076±0.000 | 0.151±0.001 | **0.306±0.003** | | CS | 0.061±0.000 | 0.081±0.001 | **0.210±0.018** |
| | BADGE | 0.054±0.001 | 0.094±0.004 | **0.193±0.013** | | BADGE | 0.060±0.001 | 0.087±0.002 | **0.201±0.020** |
| | UTILITY | 0.081±0.000 | 0.170±0.002 | **0.352±0.003** | | UTILITY | 0.065±0.000 | 0.089±0.001 | **0.325±0.017** |
| | AVERAGE | 0.057±0.009 | 0.110±0.020 | **0.241±0.043** | | AVERAGE | 0.049±0.007 | 0.068±0.010 | **0.186±0.034** |
| GTEA | CONF | 0.270±0.008 | 0.252±0.008 | **0.498±0.007** | HAPT | CONF | 0.011±0.001 | 0.019±0.001 | **0.050±0.005** |
| | ENTROPY | 0.226±0.009 | 0.220±0.010 | **0.403±0.018** | | ENTROPY | 0.009±0.000 | 0.014±0.000 | **0.033±0.002** |
| | MARG | **0.423±0.008** | 0.380±0.002 | 0.404±0.009 | | MARG | 0.022±0.001 | 0.047±0.002 | **0.148±0.007** |
| | CS | **0.437±0.014** | 0.398±0.014 | 0.371±0.014 | | CS | 0.026±0.000 | 0.048±0.002 | **0.146±0.004** |
| | BADGE | 0.348±0.014 | 0.305±0.013 | **0.404±0.008** | | BADGE | 0.026±0.000 | 0.053±0.001 | **0.160±0.008** |
| | UTILITY | **0.516±0.002** | 0.453±0.001 | 0.424±0.006 | | UTILITY | 0.028±0.000 | 0.058±0.003 | **0.257±0.004** |
| | AVERAGE | 0.370±0.041 | 0.335±0.034 | **0.418±0.016** | | AVERAGE | 0.020±0.003 | 0.040±0.007 | **0.132±0.031** |

Table 4: Classification timestamp accuracy after the final (15th) round with and without plateau width regularization and temperature scaling (the best results in bold).

| Dataset | Width Reg. | No | | Yes | | | | | | | | |
|---|---|---|---|---|---|---|---|---|---|---|---|---|
| | Temp. Scal. | No ($T=1$) | Yes ($T=2$) | $T=0.5$ | $T=0.75$ | $T=1$ | $T=1.5$ | $T=1.75$ | $T=2$ | $T=2.25$ | $T=2.5$ | $T=2.75$ |
| 50salads | CONF | 0.441±0.015 | 0.487±0.030 | 0.465±0.044 | 0.489±0.026 | 0.480±0.035 | 0.519±0.019 | 0.559±0.020 | **0.559±0.010** | 0.535±0.020 | 0.508±0.023 | 0.460±0.045 |
| | ENTROPY | 0.431±0.042 | 0.455±0.040 | 0.430±0.044 | 0.410±0.027 | 0.442±0.013 | 0.462±0.029 | 0.452±0.009 | **0.496±0.027** | 0.479±0.015 | 0.462±0.018 | 0.482±0.041 |
| | MARG | 0.655±0.033 | 0.671±0.027 | 0.668±0.033 | 0.667±0.016 | 0.691±0.025 | 0.671±0.024 | 0.682±0.018 | **0.697±0.020** | 0.664±0.028 | 0.658±0.013 | 0.599±0.030 |
| | CS | 0.611±0.028 | 0.618±0.026 | 0.592±0.040 | 0.616±0.027 | 0.624±0.034 | 0.610±0.020 | 0.627±0.029 | **0.657±0.024** | 0.656±0.018 | 0.633±0.025 | 0.626±0.015 |
| | BADGE | 0.595±0.018 | 0.599±0.020 | 0.575±0.021 | 0.594±0.037 | 0.623±0.023 | 0.631±0.017 | **0.634±0.014** | 0.600±0.025 | 0.575±0.018 | 0.546±0.028 | 0.566±0.010 |
| | UTILITY | 0.671±0.017 | 0.670±0.016 | 0.662±0.024 | 0.644±0.036 | 0.662±0.022 | 0.652±0.037 | 0.651±0.024 | **0.672±0.018** | 0.661±0.024 | 0.661±0.025 | 0.667±0.028 |
| | AVERAGE | 0.567±0.039 | 0.583±0.034 | 0.565±0.037 | 0.570±0.037 | 0.587±0.038 | 0.591±0.031 | 0.601±0.031 | **0.614±0.028** | 0.595±0.029 | 0.578±0.032 | 0.566±0.030 |
| GTEA | CONF | 0.539±0.072 | 0.575±0.052 | 0.614±0.025 | 0.587±0.020 | 0.603±0.023 | 0.607±0.010 | 0.575±0.011 | **0.654±0.011** | 0.553±0.082 | 0.638±0.027 | 0.477±0.087 |
| | ENTROPY | 0.553±0.016 | 0.578±0.020 | 0.606±0.016 | 0.574±0.018 | 0.596±0.018 | **0.614±0.019** | 0.592±0.022 | 0.590±0.021 | 0.582±0.026 | 0.551±0.022 | 0.593±0.010 |
| | MARG | 0.605±0.014 | 0.646±0.023 | 0.609±0.015 | 0.636±0.014 | 0.632±0.010 | 0.636±0.014 | 0.672±0.009 | 0.659±0.015 | **0.682±0.021** | 0.669±0.009 | 0.657±0.032 |
| | CS | 0.596±0.012 | 0.606±0.013 | 0.601±0.021 | 0.586±0.012 | 0.484±0.112 | 0.598±0.021 | 0.597±0.019 | 0.630±0.007 | 0.616±0.020 | 0.629±0.015 | **0.673±0.011** |
| | BADGE | 0.645±0.015 | 0.644±0.011 | 0.616±0.012 | 0.635±0.018 | 0.620±0.010 | 0.641±0.016 | 0.658±0.015 | 0.663±0.015 | 0.676±0.019 | **0.687±0.017** | 0.677±0.007 |
| | UTILITY | 0.589±0.043 | 0.608±0.031 | 0.605±0.011 | 0.529±0.081 | 0.616±0.016 | 0.595±0.017 | 0.627±0.017 | 0.644±0.019 | 0.638±0.014 | 0.659±0.015 | **0.667±0.013** |
| | AVERAGE | 0.588±0.014 | 0.609±0.012 | 0.608±0.002 | 0.591±0.015 | 0.592±0.020 | 0.615±0.007 | 0.620±0.014 | **0.640±0.010** | 0.624±0.019 | 0.639±0.018 | 0.624±0.029 |

## 4.4 EFFECTS OF SPARSITY-AWARE LABEL PROPAGATION TECHNIQUES IN TCLP

Table 4 shows the timestamp accuracy achieved by TCLP with and without two techniques employed to handle sparsity of labels—temperature scaling for classifier output calibration and plateau width regularization in Section 3.2.2. The class skewness balancing is employed by default to assure stable performance. In addition, the temperature scale factor $T$ is varied for the temperature scaling technique. Compared with enabling both width regularization and temperature scaling ($T = 2$), removing width regularization only, temperature scaling only, and both degrades the timestamp accuracy by 5.0%, 4.3%, and 7.6%, respectively, in the 50salads dataset and 4.8%. 4.2%, and 8.1%, respectively, in the GTEA dataset. Clearly, both techniques are helpful for the label propagation, with the width regularization showing higher effect than the temperature scaling. Note that when $T > 2$, the label propagation length is suppressed, thereby causing deficiency in labels needed to train a classifier; when $T < 1$, the label propagation length may exceed the true segment length, thereby including wrong labels when training the classifier. The break point is affected by the average length of segments in each dataset, where it occurs at a higher value of $T$ in a dataset with shorter segments: the average segment length is 34 in the GTEA dataset whereas it is 289 in the 50salads dataset.

## 5 CONCLUSION

In this paper, we present a novel label propagation framework for time-series active learning, TCLP, that fully takes advantage of the temporal coherence inherent in time-series data. The temporal coherence is modeled by the quadratic plateau model, which plays a key role in segment estimation. Furthermore, the sparsity of known labels is relieved using temperature scaling and plateau regularization. Thanks to accurate and effective label propagation, TCLP enables us to improve the performance of time-series supervised learning with much smaller labeling effort. Extensive experiments with various datasets show that TCLP improves the classification accuracy by up to 7.1 times when only 0.8% of data points are queried for their labels. Future work includes developing a query selection strategy that maximizes the merit of label propagation and utilizing a constraint on plateau model fitting based on similarity among plateau models with the same class. Overall, we expect that our work will contribute greatly to various applications whose labeling cost is expensive.

## ACKNOWLEDGEMENT

This work was partly supported by Samsung Electronics Co., Ltd. (IO201211-08051-01) through the Strategic Collaboration Academic Program and Institute of Information & Communications Technology Planning & Evaluation (IITP) grant funded by the Korea government (MSIT) (No. 2020-0-00862, DB4DL: High-Usability and Performance In-Memory Distributed DBMS for Deep Learning).

## CODE OF ETHICS

In this paper, we introduce a novel time-series active learning framework that propagates the queried labels to the neighboring data points. Our work reduces excessive effort and time spent on annotation for running deep learning system. We thereby contribute the society to run deep learning systems efficiently for personal, industrial, and social purposes. All datasets used in this paper are available in public websites and have been already anonymized, using random numeric identifiers to indicate different subjects to preserve privacy; in addition, these datasets have been extensively cited for the studies in human activity recognition and video segmentation.

## REPRODUCIBILITY

We try our best in the main text to enhance the reproducibility of our work. Section 3.3 explains the details on the procedures for the theoretical analysis. Section 4.1 clarifies the active learning setting, datasets, evaluation metrics, and configurations for training the classifier as well as the hyperparameters of our proposed method. The source code is uploaded on `https://github.com/kaist-dmlab/TCLP`, which contains the source code for active learning (main algorithm) and data preprocessing as well as detailed instructions. We download all datasets used in this paper from public websites, as specified in the corresponding references.

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

## A   COMPUTATIONAL COMPLEXITY OF TCLP

In each round of active learning, let $M$ be the number of plateau models, $V$ be the average length of sub-sequences of predicted probability $f_\theta(x_t)[k]$, and $S$ be the number of training steps for evaluating Equation (4). Then, considering constant computational complexity for calculating the loss and gradient at each timestamp in the sub-sequences, we derive the computational complexity of plateau model fitting per round to be $\mathcal{O}(MVS)$. Here, $M$ can be reduced by merging two overlapping plateau models with the same class. This complexity of fitting the plateau models is negligible compared with the complexity of training the classifier. For instance, according to the experiment for the 50salads dataset conducted using Intel Xeon Gold 6226R and Nvidia RTX3080, fitting the plateau models took only about 1 to 2 minutes, whereas training the classifier took about half an hour per active learning round.

## B   DETAILED FIGURE WITH STANDARD ERROR AND SUPERVISED ACCURACY

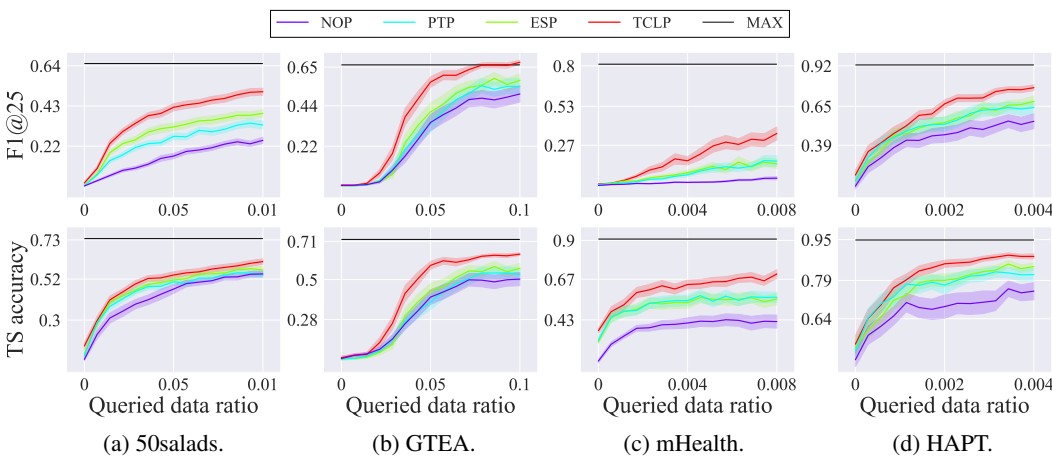

(a) 50salads.   (b) GTEA.   (c) mHealth.   (d) HAPT.

Figure 5: Classification accuracy with standard error measured at each (1st–15th) round of active learning. The accuracy value is an average over all query selection methods. The black line labeled MAX at the top indicates the maximum classification accuracy.

Figure 5 enriches Figure 4 with *standard error*, indicated by the shadow around a line, and *fully-supervised classification accuracy*, indicated by the horizontal line labeled MAX. At the last queried data ratio (i.e., after 15 active learning rounds) of each figure, in the 50salads dataset where only 1% of data points are queried, TCLP achieves 85% of the timestamp accuracy of fully-supervised classification. Similarly, in the HAPT dataset where only 0.4% of data points are queried, TCLP achieves 92% of the timestamp accuracy of fully-supervised classification. Overall, these results show that TCLP achieves the performance very close to fully-supervised classification using a very small proportion of query data points.

## C   ADJUSTMENT OF OVERLAPPING PLATEAUS IN TCLP

The process of adjusting overlapping updated plateaus through either merge or reduction is as follows. Consider two plateaus $h_{k_l}(c_l, w_l, s_l)$ (on the left) and $h_{k_r}(c_r, w_r, s_r)$ (on he right), where $(c_l - w_l < c_r - w_r) \wedge (c_l + w_l > c_r - w_r) \wedge (c_l + w_l < c_r + w_r)$ holds. If the classes assigned to these two plateaus are the same, i.e., $k_l = k_r$, then they are merged to become one plateau whose width covers the segment merged from the two plateaus' segments. Hence, the half-width $w'$ and center $c'$ of the new plateau are $w' = (c_r + w_r - c_l + w_l)/2$ and $c' = c_l - w_l + w'$, respectively. If, on the other hand, different classes are assigned to the two plateaus, i.e., $k_l \neq k_r$, their half-widths are reduced to remove any overlap between them. As a result, separating the two plateaus at the midpoint $m(= (c_l + c_r)/2)$ between their centers, after the reduction the left plateau has the half-width $w'_l = (m - (c_l - w_l))/2$ and the center at $c'_l = c_l - w_l + w'_l$, and the right plateau has the half-width $w'_r = (c_r + w_r - m)/2$ and the center $c'_r = c_r + w_r - w'_r$. Note that the labels of queried data points should not change as a result of this reduction. If the right plateau covers multiple queried data points whose timestamps are smaller than $c_r$, the timestamp of the leftmost queried data point becomes a new $c_r$. The same is applicable to the left plateau, except the change of the direction.

## D   EFFICACY OF TCLP FOR EACH QUERY SELECTION METHOD

Figures 6–11 show the efficacy of each label propagation (LP) approach combined with each query selection method. The performance plot shown in Figure 4 is the average of these results over the six query selection methods. For all query selection methods, TCLP is shown to be effective in improving active learning performance compared to the other label propagation approaches.

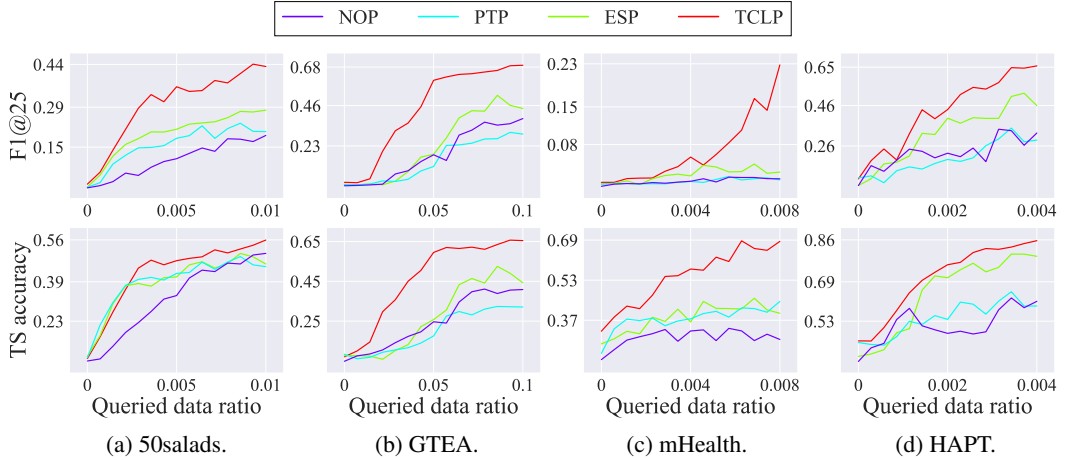

Figure 6: Efficacy of the four LP approaches with CONF query selection.

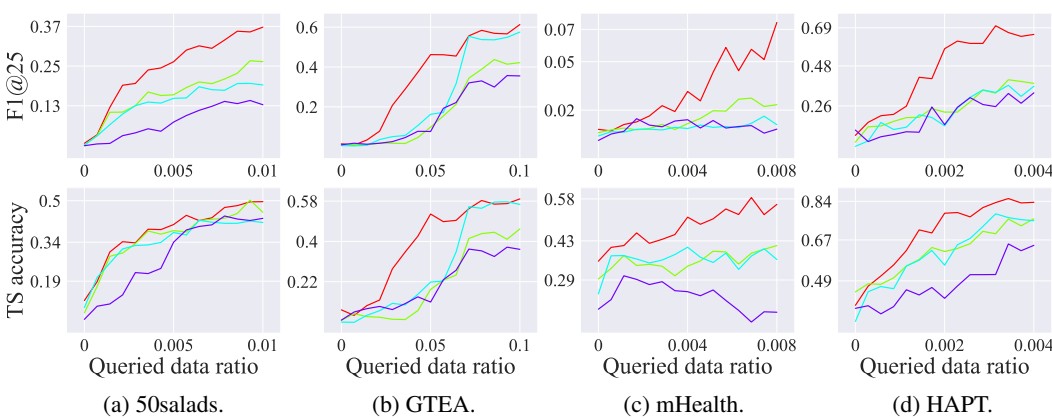

Figure 7: Efficacy of the four LP approaches with ENTROPY query selection.

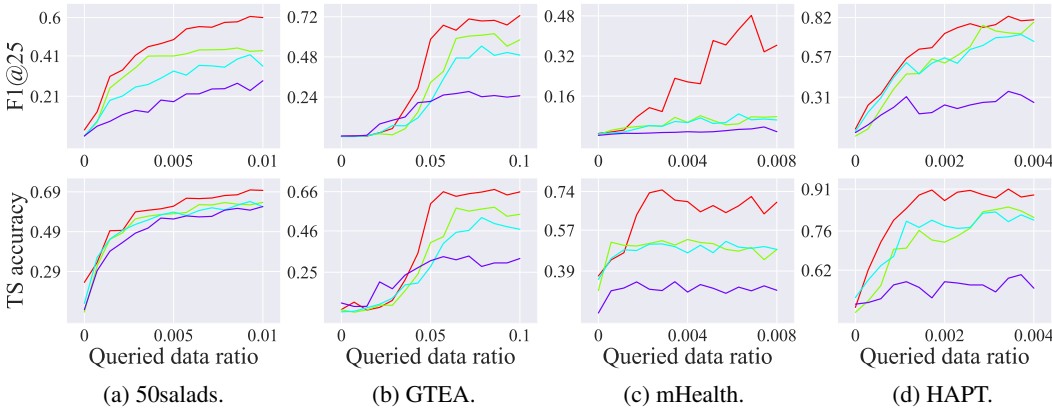

Figure 8: Efficacy of the four LP approaches with MARG query selection.

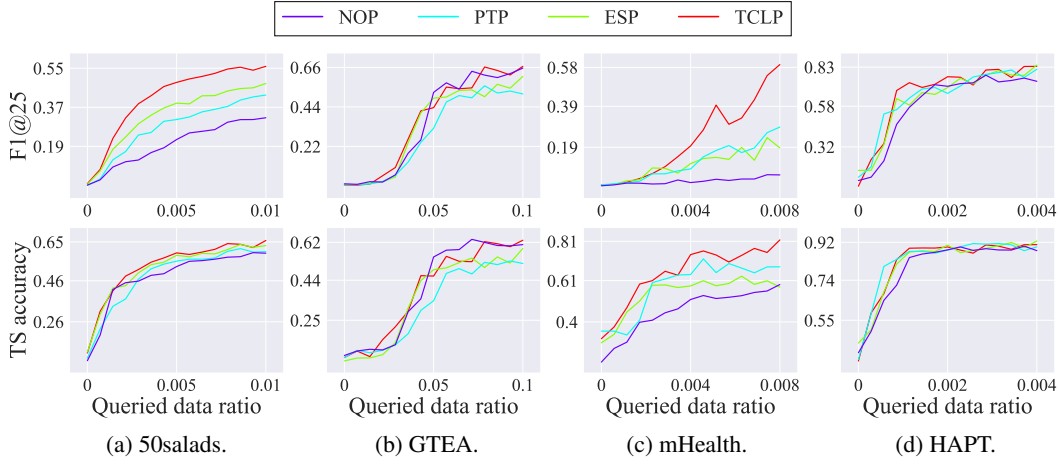

Figure 9: Efficacy of the four LP approaches with CS query selection.

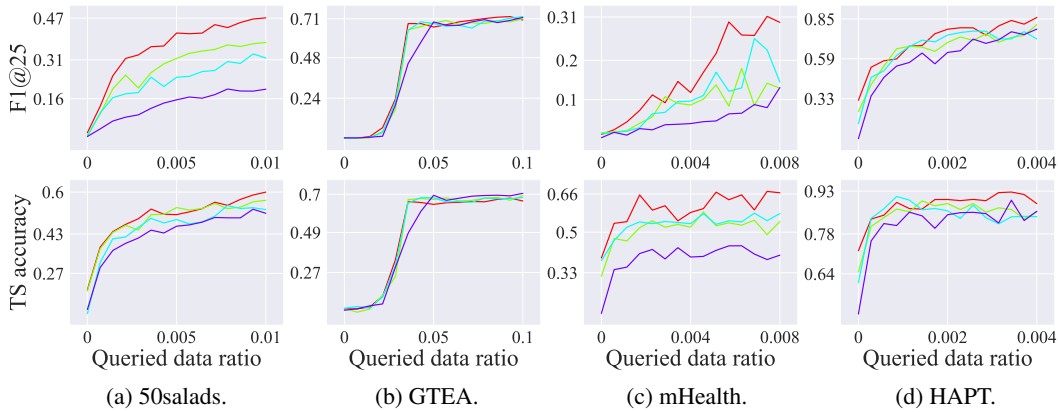

Figure 10: Efficacy of the four LP approaches with BADGE query selection.

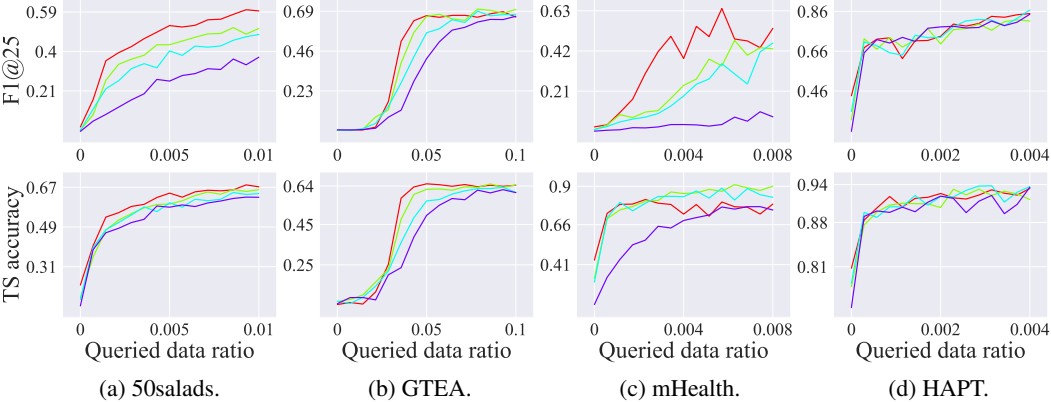

Figure 11: Efficacy of the four LP approaches with UTILITY query selection.

# E  EVALUATION OF PROPAGATED LABELS

Figures 12–17 show the propagation quality of each label propagation approach, again demonstrating the superiority of TCLP. These plots provide the overall trends including the final round results reported in Table 3. The correct propagation ratio (CPR), i.e., the number of correctly propagated labels / the total number of data points, is measured at each round using a specific query selection method on each dataset. The CPR of TCLP is shown to be higher than those of the other label propagation approaches throughout the entire period (i.e., from the 1st through the 15th round).

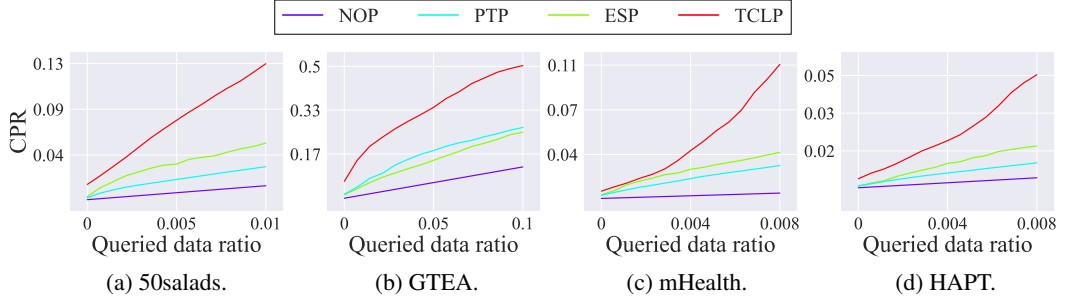

Figure 12: CPR of the four LP approaches with CONF query selection.

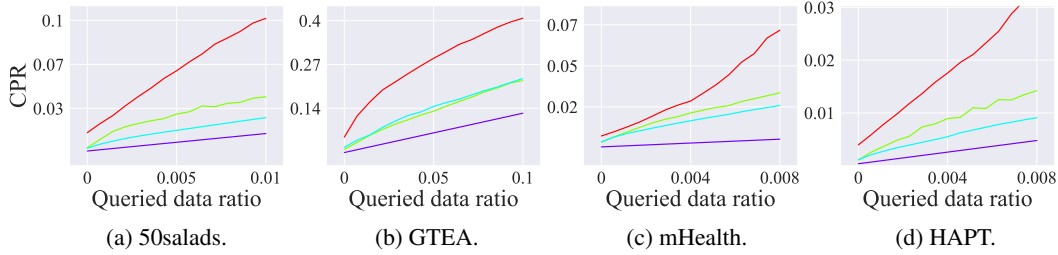

Figure 13: CPR of the four LP approaches with ENTROPY query selection.

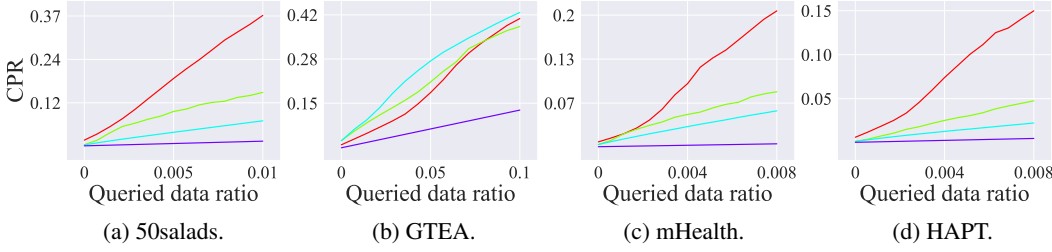

Figure 14: CPR of the four LP approaches with MARG query selection.

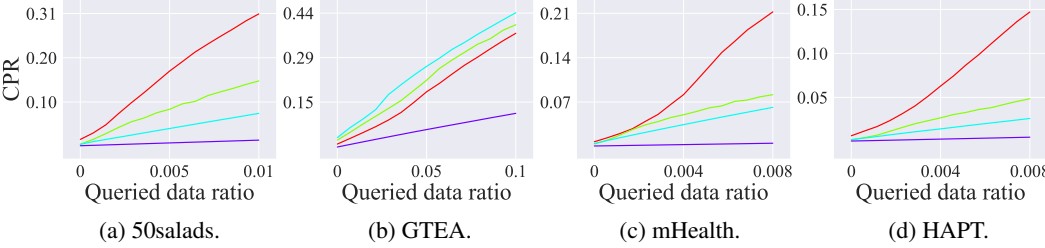

Figure 15: CPR of the four LP approaches with CS query selection.

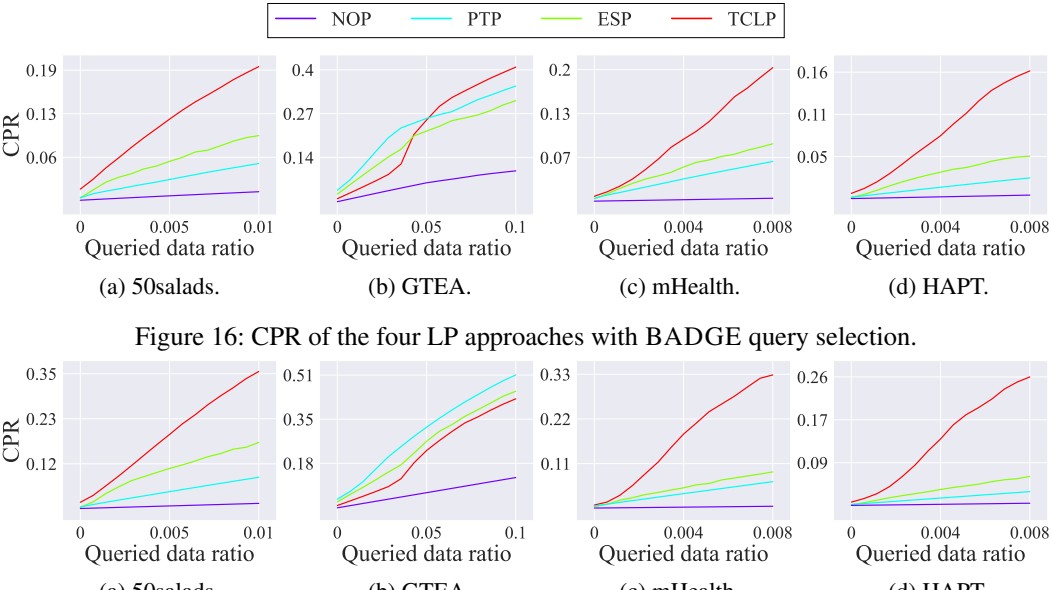

Figure 16: CPR of the four LP approaches with BADGE query selection.

Figure 17: CPR of the four LP approaches with UTILITY query selection.

