# OpenReview forum: "Coherence-based Label Propagation over Time Series for Accelerated Active Learning"
_ICLR.cc/2022/Conference — ICLR 2022 Poster_

### Official Review · Reviewer_kJBx · 2021-10-25

**Correctness:** 3
**Technical Novelty And Significance:** 2
**Empirical Novelty And Significance:** 4
**Recommendation:** 6
**Confidence:** 3

**Details Of Ethics Concerns:**

=

**Main Review:**

Strength:
- The empirical result shows that this methods is a really good method.
- The paper is a good ICLR paper, it is based on a simple idea, and it does work well in practice.

Weakness:
- Updating rules based on a learnt model is proned to overconfidence loophole. While the authors briefly discussed temperature scaling usually used to avoid overconfidence in neural networks, I would appreciate a more thorough discussion on the matter, as well as some example of failure of TCLP.
- I was not able to open the dropbox files and to read the code, but I believe the authors have reused licensed files that are not mentionned in the paper.

There is a lot of small corrections that I will detail now:

page 3:
1/ I believe D_u \cup D_l = D, but this is not mentioned clearly.

2/ "More formally, the estimated segment for t_q is Eq. (1)"
This is a choice made by the authors, and should be more clearly stated as a choice, by using “we estimate the segment for t_q”. If this choice is standard, than it would be nice to refer to litterature that introduce this, or similar, objective.

3/ What happen when the points in [t_s, t_e] are not all in D_u? For example, if segments [t_s, t_e], overlap for t_q1 and t_q2?

page 4:
1/ Eq.(4), what is t? I believe c, w, and s should be updated according to the gradient of the quantity appearing in Eq. (3) where there is a summation over t that make it disappears.

page 5:
1/ we assume that the estimated class […] is an indenpendent random variable: independent of what, where does the randomness comes from?

2/ Eq.(5) is really weird, and is based on assumptions that seems highly unrealistic. I would like to see derivations for such an equality.
It seems to me that the authors are assuming that $f_\theta(x_t)[y]$ is a random variable that is independent of t, for each label y.

3/ "Without loss of generality”, why don’t we loose generality?

Overall I believe Section 3.3 is too poor scientifically speaking to be put inside an ICLR paper.

page 6:
1/ It would be nice to describe temperature scaling in more details.
Similarly, the labels propagation from H_r is not that clear, especially if there is overlaps.

2/ In Algorithm 1, the loss ell and the query strategy should be added as input of the algorithms, as well as the stepsize lambda.

page 7:
1/ Is TCLP working better than other methods for active learning simply because it is working better for semi-supervised learning?
What happen if we start with 1% of labeled datapoints rather then nearly 0%?

2/ Can we have standard deviation on Figure 4?

3/ Is the same neural network architecture used for each methods?

page 10:
1/ Error regarding the github url.

**Summary Of The Paper:**

The goal of the paper is to improve active learning for time-series.

The goal is to learn a function from x to y, with data coming from a time series (x_t) and (y_t). It is assumed that (y_t) is piecewise constant.
At the beginning no (y_t) are known, but query can be made in a active learning framework.

This paper tries to find the pieces where (y_t) is constant based on the features (x_t). It is done using the labeled timestamps (y_t_q) at (t_q) and the current learnt model f from X to Y with a thresholding rule based on f(x_t) around the (t_q). This gives segments on which (y_t) is supposed constant egal to the (y_t_q). Those labels are added as if they were true labels. Then the model for f is updated, and some active query are made based on any model. The loop is repeated for a given number of rounds.

**Summary Of The Review:**

The experiments speak for themselves.
Overall while the idea is simple, and the math are not really satisfying, the experimental part erases most of my concerns.

---

> ### Author Response · Authors · 2021-11-20
> **Response to Reviewer kJBx**
>
> We deeply appreciate the reviewer's constructive comments and positive feedback on our manuscript.
>
> **1. While the authors briefly discussed temperature scaling usually used to avoid overconfidence in neural networks, I would appreciate a more thorough discussion on the matter, as well as some example of failure of TCLP.**
>
> Thank you for your comment.
> TCLP uses temperature scaling on estimated class distribution to avoid a wrong propagation. The temperature scaling can be summarized as follows. When $T=1$, the estimated class distribution does not change. When $T\rightarrow{}\infty$, each estimated class probability approaches $\frac{1}{K}$ (the highest entropy), where $K$ is the number of classes. When $T\rightarrow{}0$, the maximum estimated class probability becomes 1, thus showing the lowest entropy.
>
> $T$ is critical for effective label propagation in TCLP.
> In this revision, we conducted additional experiments for the 50salads and GTEA datasets, with $T$ varying in a larger range from $0.5$ to $2.75$ at the increment of $0.25$. The break point was reached at $T = 2.0$ according to the average of six query selection methods.
>
> Table 4 in the revision summarizes the augmented results. Evidently, "too low" or "too high" $T$ results in performance degradation. In the 50salads dataset, when $T > 2$, the label propagation length is suppressed, thereby causing deficiency in labels needed to train a classifier; when $T < 1$, the label propagation length may exceed the true segment length, thereby including wrong labels when training the classifier.
> The break point is affected by the average length of segments in each dataset, where it occurs at a higher value of $T$ in a dataset with shorter segments; the average segment length is 34 in the GTEA dataset whereas 289 in the 50salads dataset.
> A brief summary of this result has been added in Section 4.4 (p. 9) in the revision.
>
> **2. I was not able to open the Dropbox files and to read the code.**
>
> We apologize for any inconvenience caused by that, but the Dropbox link is working fine for us. It is possible there was a temporary network problem at that moment. Regardless, we have uploaded the source code additionally on Google Drive; it is available at [Link](https://drive.google.com/drive/folders/1_quvTIuBIMzVI4j8MkPTzFEfH-zdU-ri?usp=sharing).
>
> **3. There is a lot of small corrections that I will detail now:**
>
> **3.1. I believe $D_U \cup D_L = D$, but this is not mentioned clearly.**
>
> Thank you for pointing it out; we have added it in this revision (p. 3).
>
> **3.2. "More formally, the estimated segment for $t_q$ is Eq. (1)." This is a choice made by the authors, and should be more clearly stated as a choice, by using "we estimate the segment for $t_q$". If this choice is standard, than it would be nice to refer to literature that introduce this, or similar, objective.**
>
> It was a choice made by the authors.  We rephrased it as suggested in this revision (p. 3).
>
> **3.3. What happen when the points in $[t_s, t_e]$ are not all in $D_u$? For example, if segments $[t_s, t_e]$, overlap for $t_{q_1}$ and $t_{q_2}$?**
>
> Once data points are queried, their labels are not changed by propagation. Besides, TCLP merges overlapping plateau models when they belong to the same class. On the other hand, when two overlapping plateau models belong to different classes, we make them disjoint by reducing their widths. In this revision, we added the merge and adjustment process in Algorithm 1 (p. 6) and Appendix C.
> The details of this process are omitted because of lack of page space, and will be added in the final version.
>
> **3.4. In Eq. (4), I believe $c$, $w$, and $s$ should be updated according to the gradient of the quantity appearing in Eq. (3) where there is a summation over $t$.**
>
> It was a mistake on our end. Thank you for pointing it out. In this revision, we corrected Eq. (4) by adding a summation over $t$ (p. 4).

---

> > ### Author Response · Authors · 2021-11-20
> > **Additional Response to Reviewer kJBx**
> >
> >
> > **3.5. We assume that the estimated class $[...]$ is an independent random variable: independent of what, where does the randomness comes from?**
> >
> > Please see our response to the comment (3.6) below.
> >
> > **3.6. Eq. (5) is really weird, and is based on assumptions that seems highly unrealistic. It seems to me that the authors are assuming that $f_\theta(x_t)[k]$ is a random variable that is independent of $t$, for each label $y$.**
> >
> >
> > In the manuscript we omitted some details in the assumption, and apparently it made the assumption look unrealistic. Our assumption is based on a widely-used sequence classification framework (Graves et al., 2006;Chung et al., 2015). In $p(\pi|{x})=\prod_{t=1}^{T}y_{\pi_t}^{t}$ from Graves et al. (2006), the estimated probability at each timestamp in a segment is multiplied to obtain the overall segment probability. The underlying assumption of this equation is that the estimated probability at a timestamp (denoted as $f_\theta(x_t)[k]$ in our paper) is conditionally independent of the estimated probabilities at other timestamps in a given input sequence. We apply this assumption to our problem to derive Eq. (5) in our paper.
> >
> > That is, we also multiply $z=Pr(f_\theta(x_t)[k]>\delta$ for $l$ times and $(1-z)^2$ to guarantee that the condition $f_\theta(x_t)[k]>\delta$ is satisfied for $l$ consecutive timestamps. In addition, the query timestamp $t_q$ can be placed at  $l$ different locations in the sequence of a length $l$. Therefore, the probability of an estimated segment reaching a length $l$ is $l\cdot z^{l-1}(1-z)^2$.
> > We clarified this assumption in Section 3.3 and will further improve the presentation of Section 3.3 in the final version.
> >
> > **3.7. "Without loss of generality" why don't we loose generality?**
> >
> > We removed this phrase in this revision in order to avoid any confusion (p. 5).
> >
> > **3.8. It would be nice to describe temperature scaling in more details. Similarly, the label propagation from $H_r$ is not that clear, especially if there is an overlap.**
> >
> > Please see our responses to the comments (1) and (3.3).
> >
> > **3.9. In Algorithm 1, the loss $\ell$ and the query strategy should be added as the input of the algorithm, as well as the step size $\lambda$.**
> >
> > We have added them as the input of the algorithm in this revision (p. 6).
> >
> > **3.10. Is TCLP working better than other methods for active learning simply because it is working better for semi-supervised learning? What happen if we start with 1% of labeled data points rather then nearly 0%?**
> >
> > If we start with 1% of labeled data points, the initial accuracy will be higher (than if we start with 0%), and thus, it is expected that the accuracy of TCLP will approach that of fully-supervised classification in \emph{fewer} active learning rounds.
> >
> > **3.11. Can we have standard deviation on Figure 4?**
> >
> > In this revision, we have added another figure, Figure 5 in Appendix B, which shows standard error.
> >
> > **3.12. Is the same neural network architecture used for each method?**
> >
> > Yes, we used the same neural network architecture, MS-TCN (Farha & Gall, 2019), throughout all experiments. The MS-TCN is mentioned in the paragraph "TCLP implementation details" in Section 4.1 of the manuscript (p. 7).
> >
> > **3.13. Error regarding the GitHub URL.**
> >
> > Please see our response to the comment (2) above.
> >
> >
> > **REFERENCES**
> > 1. Junyoung Chung, Kyle Kastner, Laurent Dinh, Kratarth Goel, Aaron C Courville, and Yoshua Bengio.  A recurrent latent variable model for sequential data.  In Advances in Neural Information Processing Systems 28: Annual Conference on Neural Information Processing Systems (NIPS), pp. 2980-2988, December 2015.
> > 2. Alex Graves, Santiago Fern ́andez, Faustino Gomez, and J ̈urgen Schmidhuber.  Connectionist temporal classification: Labelling unsegmented sequence data with recurrent neural networks. In Proceedings of the 23rd International Conference on Machine Learning (ICML), pp. 369-376, June 2006.
> > 3. Yazan Abu Farha and Jurgen Gall. MS-TCN: Multi-stage temporal convolutional network for action segmentation. In Proceedings of the 2019 IEEE/CVF Conference on Computer Vision and Pattern Recognition (CVPR), pp. 3575-3584, June 2019.

---

> > > ### Author Response · Authors · 2021-11-20
> > > **Additional Response to Reviewer kJBx**
> > >
> > > | Dataset  | Width Reg. and Temp. Scal. | No, No($T=1$)     | No, Yes($T=2$)     | Yes, $T=0.5$      | Yes, $T=0.75$     | Yes, $T=1$        | Yes, $T=1.5$          | Yes, $T=1.75$         | Yes, $T=2$            | Yes, $T=2.25$         | Yes, $T=2.5$          | Yes, , $T=2.75$       |
> > > |----------|----------------------------|-------------------|--------------------|-------------------|-------------------|-------------------|-----------------------|-----------------------|-----------------------|-----------------------|-----------------------|-----------------------|
> > > | 50salads |  CONF                      |  0.441$\pm$0.015  |  0.487$\pm$0.030   |  0.465$\pm$0.044  |  0.489$\pm$0.026  |  0.480$\pm$0.035  |  0.519$\pm$0.019      |  0.559$\pm$0.020      |  **0.559$\pm$0.010**  |  0.535$\pm$0.020      |  0.508$\pm$0.023      |  0.460$\pm$0.045      |
> > > | 50salads |  ENTROPY                   |  0.431$\pm$0.042  |  0.455$\pm$0.040   |  0.430$\pm$0.044  |  0.410$\pm$0.027  |  0.442$\pm$0.013  |  0.462$\pm$0.029      |  0.452$\pm$0.009      |  **0.496$\pm$0.027**  |  0.479$\pm$0.015      |  0.462$\pm$0.018      |  0.482$\pm$0.041      |
> > > | 50salads |  MARG                      |  0.655$\pm$0.033  |  0.671$\pm$0.027   |  0.668$\pm$0.033  |  0.667$\pm$0.016  |  0.691$\pm$0.025  |  0.671$\pm$0.024      |  0.682$\pm$0.018      |  **0.697$\pm$0.020**  |  0.664$\pm$0.028      |  0.658$\pm$0.013      |  0.599$\pm$0.030      |
> > > | 50salads |  CS                        |  0.611$\pm$0.028  |  0.618$\pm$0.026   |  0.592$\pm$0.040  |  0.616$\pm$0.027  |  0.624$\pm$0.034  |  0.610$\pm$0.020      |  0.627$\pm$0.029      |  **0.657$\pm$0.024**  |  0.656$\pm$0.018      |  0.633$\pm$0.025      |  0.626$\pm$0.015      |
> > > | 50salads |  BADGE                     |  0.595$\pm$0.018  |  0.599$\pm$0.020   |  0.575$\pm$0.021  |  0.594$\pm$0.037  |  0.623$\pm$0.023  |  0.631$\pm$0.017      |  **0.634$\pm$0.014**  |  0.600$\pm$0.025      |  0.575$\pm$0.018      |  0.546$\pm$0.028      |  0.566$\pm$0.010      |
> > > | 50salads |  UTILITY                   |  0.671$\pm$0.017  |  0.670$\pm$0.016   |  0.662$\pm$0.024  |  0.644$\pm$0.036  |  0.662$\pm$0.022  |  0.652$\pm$0.037      |  0.651$\pm$0.024      |  **0.672$\pm$0.018**  |  0.661$\pm$0.024      |  0.661$\pm$0.025      |  0.667$\pm$0.028      |
> > > | 50salads |  AVERAGE                   |  0.567$\pm$0.039  |  0.583$\pm$0.034   |  0.565$\pm$0.037  |  0.570$\pm$0.037  |  0.587$\pm$0.038  |  0.591$\pm$0.031      |  0.601$\pm$0.031      |  **0.614$\pm$0.028**  |  0.595$\pm$0.029      |  0.578$\pm$0.032      |  0.566$\pm$0.030      |
> > > | GTEA     |  CONF                      |  0.539$\pm$0.072  |  0.575$\pm$0.052   |  0.614$\pm$0.025  |  0.587$\pm$0.020  |  0.603$\pm$0.023  |  0.607$\pm$0.010      |  0.575$\pm$0.011      |  **0.654$\pm$0.011**  |  0.553$\pm$0.082      |  0.638$\pm$0.027      |  0.477$\pm$0.087      |
> > > | GTEA     |  ENTROPY                   |  0.553$\pm$0.016  |  0.578$\pm$0.020   |  0.606$\pm$0.016  |  0.574$\pm$0.018  |  0.596$\pm$0.018  |  **0.614$\pm$0.019**  |  0.592$\pm$0.022      |  0.590$\pm$0.021      |  0.582$\pm$0.026      |  0.551$\pm$0.022      |  0.593$\pm$0.010      |
> > > | GTEA     |  MARG                      |  0.605$\pm$0.014  |  0.646$\pm$0.023   |  0.609$\pm$0.015  |  0.636$\pm$0.014  |  0.632$\pm$0.010  |  0.636$\pm$0.014      |  0.672$\pm$0.009      |  0.659$\pm$0.015      |  **0.682$\pm$0.021**  |  0.669$\pm$0.009      |  0.657$\pm$0.032      |
> > > | GTEA     |  CS                        |  0.596$\pm$0.012  |  0.606$\pm$0.013   |  0.601$\pm$0.021  |  0.586$\pm$0.012  |  0.484$\pm$0.112  |  0.598$\pm$0.021      |  0.597$\pm$0.019      |  0.630$\pm$0.007      |  0.616$\pm$0.020      |  0.629$\pm$0.015      |  **0.673$\pm$0.011**  |
> > > | GTEA     |  BADGE                     |  0.645$\pm$0.015  |  0.644$\pm$0.011   |  0.616$\pm$0.012  |  0.635$\pm$0.018  |  0.620$\pm$0.010  |  0.641$\pm$0.016      |  0.658$\pm$0.015      |  0.663$\pm$0.015      |  0.676$\pm$0.019      |  **0.687$\pm$0.017**  |  0.677$\pm$0.007      |
> > > | GTEA     |  UTILITY                   |  0.589$\pm$0.043  |  0.608$\pm$0.031   |  0.605$\pm$0.011  |  0.529$\pm$0.081  |  0.616$\pm$0.016  |  0.595$\pm$0.017      |  0.627$\pm$0.017      |  0.644$\pm$0.019      |  0.638$\pm$0.014      |  0.659$\pm$0.015      |  **0.667$\pm$0.013**  |
> > > | GTEA     |  AVERAGE                   |  0.588$\pm$0.014  |  0.609$\pm$0.012   |  0.608$\pm$0.002  |  0.591$\pm$0.015  |  0.592$\pm$0.020  |  0.615$\pm$0.007      |  0.620$\pm$0.014      |  **0.640$\pm$0.010**  |  0.624$\pm$0.019      |  0.639$\pm$0.018      |  0.624$\pm$0.029      |

---

> > > > ### Comment · Reviewer_kJBx · 2021-11-21
> > > > **Discussion**
> > > >
> > > > I appreciate the authors answer.
> > > > However, there is few points not answered.
> > > > When I talk about the error regarding Github url I was talking about the fact that the work of Abu Farha is put with the paper of Fathi et al.
> > > >
> > > > The authors have not answered my question about licensing issues.
> > > >
> > > > I am still unclear about section 3.3.
> > > > You talked about conditional independence given x. Given x, I believe the prediction are deterministic. What am I missing?
> > > > Also in the manuscript you removed the fact that you assume independence, making this section really not satisfying in my point of view.
> > > >
> > > > Also, my question regarding the fact that the benefit might be due to the semi-supervised learning and not the active learning one, was not answered.

---

> > > > > ### Comment · Reviewer_kJBx · 2021-11-23
> > > > > **Breaking anonymity**
> > > > >
> > > > > Sadly, the google drive repository with the code contains the name of the author.
> > > > > This breaks the anonymous reviewing process!

---

> > > > > > ### Author Response · Authors · 2021-11-23
> > > > > > **Fixed the anonymity issue**
> > > > > >
> > > > > > We immediately deleted the problematic Google Drive link and will replace it with a new link.  **Please acknowledge that the Dropbox link included in the manuscript is properly anonymized.**  The Google Drive link was just created to provide you with another way for accessing our source code during the rebuttal process.  We, unfortunately, made a minor mistake in configuring the Google Drive link.  We hope that you kindly understand this situation.  Thank you again for your careful review.

---

> > > > > > > ### Comment · Reviewer_kJBx · 2021-11-23
> > > > > > > **No worries!**
> > > > > > >
> > > > > > > No worries, those things happened, it is clearly a minor point.
> > > > > > > Sorry, I was not able to access dropbox in the past and I caused you extra work.

---

> > > > > ### Author Response · Authors · 2021-11-23
> > > > > **Additional Response to Reviewer kJBx**
> > > > >
> > > > > **1. When I talk about the error regarding Github url, I was talking about the fact that the work of Abu Farha is put with the paper of Fathi et al.**
> > > > >
> > > > > Thanks for clarifying your comment. We have corrected the GitHub URL in the revision (p. 11-13).
> > > > >
> > > > > **2. The authors have not answered my question about licensing issues.**
> > > > >
> > > > > We apologize having missed this question. We have double checked that no licensed file is used in our source code. We implemented TCLP and all the other algorithms on our own to avoid unfair comparison caused by different implementation schemes. In addition, we used the API of several open-source libraries such as Tensorflow, Scipy, and Scikit-learn. Overall, there is no license issue in our implementation.
> > > > >
> > > > > **3. I am still unclear about section 3.3. You talked about conditional independence given x. Given x, I believe the prediction are deterministic. What am I missing? Also in the manuscript you removed the fact that you assume independence, making this section really not satisfying in my point of view.**
> > > > >
> > > > > Thank you so much for your careful review.
> > > > >
> > > > > Let us elaborate further on our answer.
> > > > > The output $Y$ of the classifier is typically deterministic given an input $X$ and a fixed classifier $f$. In such a circumstance where $Y= \{ f_\theta(x_1)[k],...,f_\theta(x_L)[k]) \}$ for $X$ is given, we assume that the output probabilities $f_\theta(x_t)[k]$ $(1 \leq t \leq L)$ in $Y$ are conditionally independent of one another, following several relevant literature (e.g., Graves et al. (2006)). To the best of our knowledge, the reasoning on this assumption is that $f_\theta(x_i)[k]$ is not directly used to derive $f_\theta(x_j)[k]$ where $i \neq j$.
> > > > >
> > > > > > For your reference, let us quote the relevant paragraphs from Graves et al. (2006): "Eq. (2) $p(\pi|x)=\prod_{t=1}^{T}y_{\pi_t}^{t}$.
> > > > > > Implicit in (2) is the assumption that the network outputs at different times are conditionally independent, given the internal state of the network. This is ensured by requiring that no feedback connections exist from the output layer to itself or the network."
> > > > >
> > > > > We hope our answer and further revision of the manuscript resolved your remaining concern.
> > > > >
> > > > > **4. Also, my question regarding the fact that the benefit might be due to the semi-supervised learning and not the active learning one, was not answered.**
> > > > >
> > > > > If you consider label propagation as semi-supervised learning, the benefit is largely due to semi-supervised learning, because propagated labels are much more than queried labels (e.g., more than 40 times). As clarified in our paper, TCLP can boost the performance of an off-the-shelf active learning strategy, thanks to a larger number of correctly propagated labels.

---

### Official Review · Reviewer_pVjg · 2021-11-01

**Correctness:** 4
**Technical Novelty And Significance:** 3
**Empirical Novelty And Significance:** 4
**Recommendation:** 6
**Confidence:** 3

**Main Review:**

Strengths: Applying plateau model (Equation 2) into time-series active learning for pseudo labeling is a good idea for promoting the performance. The paper is easy to follow, the idea is simple and the experimental results show significant improvement (considering both performance gain and label reducing) compare with other label propagation methods.

Remaining questions:
1) In Table 1, why b and w0 are inconsistent in the experiments across different datasets? Does it make any difference to have a larger b? That is, does the performance of TCLP decrease as the query size increases? Since in active learning, enlarging the batch is an effective way to reduce the number of rounds.
2) For temperature scaling parameter T, why the larger of T, the better the performance are? It's better to include more settings of T, i.e., T < 1, T > 2 and more analysis about the influence of T, since the value of T seems to have a significant impact on performance.
3) Besides UTILITY, it's better to add the evaluation results that trained on whole datasets as comparison.

**Summary Of The Paper:**

This paper addresses the label propagation segment estimation problem in time-series active learning, and apply plateau function to model temporal coherence. The experimental results show their effectiveness compare with baseline label propagation approaches.

**Summary Of The Review:**

I vote for a weak ac of the current version since the idea of using plateau model for pseudo labeling in time-series AL is interesting. Although there is no rigorous theory to tell us why TCLP improves performance (at least in this paper), that is, why plateau model could provide a better pseudo labeling, and what's the upper limit of its promotion. The experimental results are fairly good, they demonstrate TCLP's effectiveness in their experiments.

---

> ### Author Response · Authors · 2021-11-20
> **Response to Reviewer pVjg**
>
> We deeply appreciate the reviewer's constructive comments and positive feedback on our manuscript.
>
> **1. There is no rigorous theory to tell us why TCLP improves performance, i.e., why a plateau model could provide a better pseudo labeling and what's the upper limit of its promotion.**
>
> In Section 3.3, we show that a plateau-based segment estimation (i.e., TCLP) produces longer and more accurate segments than a threshold-based segment estimation. This is because the plateau-based approach is aware of the coherence in continuous data points, but the threshold-based approach is not, just considering an instantaneous data point. More specifically, the plateau model is *robust to outliers*, such as rapid probability fluctuations, in estimated class probabilities. These outliers are fairly common in temporal classification tasks (e.g., action classification) and referred to as over-segmentation error (Wang et al., 2020; Ahn & Lee, 2021).
> The upper limit of TCLP's promotion seems to be hard to analyze because it is dependent on the capability of a trained classifier, and we will explore this theoretical analysis in the future work.
>
> **2. Why $b$ and $w_0$ are inconsistent across datasets? Does it make any difference to have a larger $b$?**
>
> When the query size $b$ increases, the benefit of TCLP increases as well, but the rate of increase will decrease as a larger number of fitted plateau models overlap with each other.  In this regard, we agree that the value of $b$ should better be consistent across the datasets. Thus, we re-ran the experiment for the mHealth dataset with $b=200$ as well. The results newly obtained are similar to those previously obtained with $b=85$; TCLP again significantly outperforms all other methods. In the revision, the new results are reflected in Table 2, Table 3, and Figure 4, as well as the texts in Sections 4.2 and 4.3. $w_0$ is a bit different,
> because it should better be tailored to the dataset for accurate label propagation when the plateau models are initialized. Our experience indicates that any value of $w_0$ smaller than 20% of the mean segment length is adequate enough for TCLP to work well.
>
> **3-1. For the temperature scaling parameter $T$, why does a larger $T$ achieve a better performance?**
>
> For a sufficiently large $T$, temperature scaling can inhibit excessive expansion of plateau models, i.e. excessive increase in $w$, leading to correct label propagation and lower class imbalance in propagated labels.
>
> **3-2. It's better to include more settings of $T$, i.e., $T < 1$, $T > 2$ and more analysis about the influence of $T$.**
>
> In this revision, we conducted additional experiments for the 50salads and GTEA datasets, with $T$ varying in a larger range from $0.5$ to $2.75$ at the increment of $0.25$. The break point was reached at $T = 2.0$ according to the average of six query selection methods. We enriched the content of Table 4 with the additional results and discussed it in Section 4.4 (p. 9).
>
> As shown in the new Table 4, evidently "too low" or "too high" $T$ results in performance degradation. In the 50salads dataset, when $T > 2$, the label propagation length is suppressed, thereby causing deficiency in labels needed to train a classifier; when $T < 1$, the label propagation length may exceed the true segment length, thereby including wrong labels when training the classifier.
>
> The break point is affected by the average length of segments in each dataset, where it occurs at a higher value of $T$ in a dataset with shorter segments: the average segment length is 34 in the GTEA dataset whereas it is 289 in the 50salads dataset.
>
> **4. Besides UTILITY, it's better to add the evaluation results trained on whole datasets as comparison.**
>
> In the revision, we added a figure (Figure 5 in Appendix B) that shows the accuracy of fully-supervised classification (i.e., the horizontal line MAX).
>
> After 15 active learning rounds, in the 50salads dataset where only 1% of data points were queried, TCLP achieved 85% of the timestamp accuracy of fully-supervised classification. Similarly, in the HAPT dataset where only 0.4% of data points were queried, TCLP achieved 92% of the timestamp accuracy of fully-supervised classification. Overall, these results show that TCLP achieves the performance very close to fully-supervised classification using a very small proportion of query data points. This paragraph was added to Appendix B in this revision.
>
> **REFERENCES**
> 1. Hyemin Ahn and Dongheui Lee.  Refining action segmentation with hierarchical video representations. In Proceedings of the 2021 IEEE/CVF International Conference on Computer Vision (ICCV), pp. 16302-16310, October 2021.
> 2. Zhenzhi Wang, Ziteng Gao, Limin Wang, Zhifeng Li, and Gangshan Wu. Boundary-aware cascade networks for temporal action segmentation. In Proceedings of the 2020 European Conference on Computer Vision (ECCV), pp. 34-51, August 2020.

---

> > ### Author Response · Authors · 2021-11-20
> > **Additional Response to Reviewer pVjg**
> >
> > | Dataset  | Width Reg. and Temp. Scal. | No, No($T=1$)     | No, Yes($T=2$)     | Yes, $T=0.5$      | Yes, $T=0.75$     | Yes, $T=1$        | Yes, $T=1.5$          | Yes, $T=1.75$         | Yes, $T=2$            | Yes, $T=2.25$         | Yes, $T=2.5$          | Yes, , $T=2.75$       |
> > |----------|----------------------------|-------------------|--------------------|-------------------|-------------------|-------------------|-----------------------|-----------------------|-----------------------|-----------------------|-----------------------|-----------------------|
> > | 50salads |  CONF                      |  0.441$\pm$0.015  |  0.487$\pm$0.030   |  0.465$\pm$0.044  |  0.489$\pm$0.026  |  0.480$\pm$0.035  |  0.519$\pm$0.019      |  0.559$\pm$0.020      |  **0.559$\pm$0.010**  |  0.535$\pm$0.020      |  0.508$\pm$0.023      |  0.460$\pm$0.045      |
> > | 50salads |  ENTROPY                   |  0.431$\pm$0.042  |  0.455$\pm$0.040   |  0.430$\pm$0.044  |  0.410$\pm$0.027  |  0.442$\pm$0.013  |  0.462$\pm$0.029      |  0.452$\pm$0.009      |  **0.496$\pm$0.027**  |  0.479$\pm$0.015      |  0.462$\pm$0.018      |  0.482$\pm$0.041      |
> > | 50salads |  MARG                      |  0.655$\pm$0.033  |  0.671$\pm$0.027   |  0.668$\pm$0.033  |  0.667$\pm$0.016  |  0.691$\pm$0.025  |  0.671$\pm$0.024      |  0.682$\pm$0.018      |  **0.697$\pm$0.020**  |  0.664$\pm$0.028      |  0.658$\pm$0.013      |  0.599$\pm$0.030      |
> > | 50salads |  CS                        |  0.611$\pm$0.028  |  0.618$\pm$0.026   |  0.592$\pm$0.040  |  0.616$\pm$0.027  |  0.624$\pm$0.034  |  0.610$\pm$0.020      |  0.627$\pm$0.029      |  **0.657$\pm$0.024**  |  0.656$\pm$0.018      |  0.633$\pm$0.025      |  0.626$\pm$0.015      |
> > | 50salads |  BADGE                     |  0.595$\pm$0.018  |  0.599$\pm$0.020   |  0.575$\pm$0.021  |  0.594$\pm$0.037  |  0.623$\pm$0.023  |  0.631$\pm$0.017      |  **0.634$\pm$0.014**  |  0.600$\pm$0.025      |  0.575$\pm$0.018      |  0.546$\pm$0.028      |  0.566$\pm$0.010      |
> > | 50salads |  UTILITY                   |  0.671$\pm$0.017  |  0.670$\pm$0.016   |  0.662$\pm$0.024  |  0.644$\pm$0.036  |  0.662$\pm$0.022  |  0.652$\pm$0.037      |  0.651$\pm$0.024      |  **0.672$\pm$0.018**  |  0.661$\pm$0.024      |  0.661$\pm$0.025      |  0.667$\pm$0.028      |
> > | 50salads |  AVERAGE                   |  0.567$\pm$0.039  |  0.583$\pm$0.034   |  0.565$\pm$0.037  |  0.570$\pm$0.037  |  0.587$\pm$0.038  |  0.591$\pm$0.031      |  0.601$\pm$0.031      |  **0.614$\pm$0.028**  |  0.595$\pm$0.029      |  0.578$\pm$0.032      |  0.566$\pm$0.030      |
> > | GTEA     |  CONF                      |  0.539$\pm$0.072  |  0.575$\pm$0.052   |  0.614$\pm$0.025  |  0.587$\pm$0.020  |  0.603$\pm$0.023  |  0.607$\pm$0.010      |  0.575$\pm$0.011      |  **0.654$\pm$0.011**  |  0.553$\pm$0.082      |  0.638$\pm$0.027      |  0.477$\pm$0.087      |
> > | GTEA     |  ENTROPY                   |  0.553$\pm$0.016  |  0.578$\pm$0.020   |  0.606$\pm$0.016  |  0.574$\pm$0.018  |  0.596$\pm$0.018  |  **0.614$\pm$0.019**  |  0.592$\pm$0.022      |  0.590$\pm$0.021      |  0.582$\pm$0.026      |  0.551$\pm$0.022      |  0.593$\pm$0.010      |
> > | GTEA     |  MARG                      |  0.605$\pm$0.014  |  0.646$\pm$0.023   |  0.609$\pm$0.015  |  0.636$\pm$0.014  |  0.632$\pm$0.010  |  0.636$\pm$0.014      |  0.672$\pm$0.009      |  0.659$\pm$0.015      |  **0.682$\pm$0.021**  |  0.669$\pm$0.009      |  0.657$\pm$0.032      |
> > | GTEA     |  CS                        |  0.596$\pm$0.012  |  0.606$\pm$0.013   |  0.601$\pm$0.021  |  0.586$\pm$0.012  |  0.484$\pm$0.112  |  0.598$\pm$0.021      |  0.597$\pm$0.019      |  0.630$\pm$0.007      |  0.616$\pm$0.020      |  0.629$\pm$0.015      |  **0.673$\pm$0.011**  |
> > | GTEA     |  BADGE                     |  0.645$\pm$0.015  |  0.644$\pm$0.011   |  0.616$\pm$0.012  |  0.635$\pm$0.018  |  0.620$\pm$0.010  |  0.641$\pm$0.016      |  0.658$\pm$0.015      |  0.663$\pm$0.015      |  0.676$\pm$0.019      |  **0.687$\pm$0.017**  |  0.677$\pm$0.007      |
> > | GTEA     |  UTILITY                   |  0.589$\pm$0.043  |  0.608$\pm$0.031   |  0.605$\pm$0.011  |  0.529$\pm$0.081  |  0.616$\pm$0.016  |  0.595$\pm$0.017      |  0.627$\pm$0.017      |  0.644$\pm$0.019      |  0.638$\pm$0.014      |  0.659$\pm$0.015      |  **0.667$\pm$0.013**  |
> > | GTEA     |  AVERAGE                   |  0.588$\pm$0.014  |  0.609$\pm$0.012   |  0.608$\pm$0.002  |  0.591$\pm$0.015  |  0.592$\pm$0.020  |  0.615$\pm$0.007      |  0.620$\pm$0.014      |  **0.640$\pm$0.010**  |  0.624$\pm$0.019      |  0.639$\pm$0.018      |  0.624$\pm$0.029      |

---

### Official Review · Reviewer_s4ST · 2021-11-01

**Correctness:** 3
**Technical Novelty And Significance:** 3
**Empirical Novelty And Significance:** 3
**Recommendation:** 6
**Confidence:** 2

**Main Review:**

Pros:
1. Good approach to use limited user feedback efficiently.
2. Satisfactory set of experiments

Cons:
1. Need some more elaboration on the theory
2. Computational/time complexity is missing


Main Comments:

1. Equation 3: It seems like c, w, s are all estimated independently for each plateau. However, we know that very likely, these will be similar for a particular y_l (class label). Could we use such a constraint/model? E.g., we might have a [parameterized] model for (c_l, w_l, s_l) which relates to y_l.

2. Equation 5: This expression needs to be elaborated and also needs to be shown that the sum of all Pr(t_e - t_s = i) where i = 1, ..., L is 1 so that we can claim Pr is a valid probability. This is not obvious.

3. Algorithm 1: Do we always have r*b plateau models at the r-th round of active learning? What is the computational complexity and/or time for each round? It seems that computation time would increase with each round. Is there a way to lower the complexity by taking a sample of the most relevant plateau models?

4. It is unclear why 'b' is 85 for mHealth (85 for mHealth vs 200 for others) -- no reasoning has been provided. To make sure that the settings were not selected after-the-fact just to get the best results (in table 2), the same settings should be used across all datasets unless there is a really good justification. I assume w_0 is diferent for GTEA (5 for GTEA vs 15 for others) since it has the smallest segments. It would be better if these parameters could be auto-selected or the paper explicitly stated some reasoning/rule to select appropriate values for them.

**Summary Of The Paper:**

The paper presents an active learning algorithm for classifying and segmenting timeseries data. When instances are labeled by users, these labels are propagated to [temporally] neighboring instances using a plateau model (if possible) so that the information is shared more efficiently across multiple instances.

**Summary Of The Review:**

This paper proposes a technique to improve accuracy in an important field of application where data volumes are high and efficient use of limited user feedback is of urgent need. The plateau model makes intuitive sense in the context employed here and looks like something we would want to add to the toolbox.

---

> ### Author Response · Authors · 2021-11-20
> **Response to Reviewer s4ST**
>
> We deeply appreciate the reviewer's constructive comments and positive feedback on our manuscript.
>
> **1. Can we use a constraint on plateau model fitting using similarity among plateau models with the same class?**
>
> This is an excellent suggestion.  Indeed, in the 50salads video dataset,  similar actions (e.g., cutting a tomato) tend to have similar lengths. Thus, we can use the average action length estimated from the learned plateaus as the initial value of $w_k$ after querying a data point with the label $k$. One simple way to implement this idea is to average the values of $w_k$ from the previous rounds to generate the initial $w_k$ of the plateau at the current round. In this revision, we have added this idea as part of the future work.
>
> **2. Can we show that Eq. (5) is a valid probability by proving that Eq. (5) sums to 1?**
>
> Thank you very much for your careful review. Let us first clarify that Eq. (5) has nothing to do with the plateau-based approach proposed in TCLP but is only for the threshold-based approach, which was introduced as a baseline for theoretical comparison with the plateau-based approach. So, our intent in using Eq. (5) is to prove that the threshold-based approach cannot estimate the true segment as accurately as the plateau-based approach can, especially when the segment's length is significantly large.
>
> Let us now provide the proof to clarify our point.
>
> Sum of Eq. (5) over all possible length $l = 1,...,\infty$ of an estimated segment is derived to be equal to 1 as shown below:
>
> $\sum_{l=1}^{\infty}l \cdot z^{l-1}(1-z)^2 = \lim_{L \to \infty} \frac{(1-z)^2}{z} \sum_{l=1}^{L}l \cdot z^{l}$
>
> $= \lim_{L \to \infty} \frac{(1-z)^2}{z} \cdot z \cdot \frac{1-(L+1)z^{L}+Lz^{L+1}}{(1-z)^2} \text{  (by the power series formula)}$
>
> $= \lim_{L \to \infty} 1-(L+1)z^{L}+Lz^{L+1}$
>
> $=1$,
>
> where $0 < z < 1$. We believe this confirms that Eq. (5) is a valid probability.
>
> **3-1. Do we always have $r\times b$ plateau models at the $r$-th round?**
>
> The number of plateau models can be less than $r\times b$ at the $r$-th round as we merge two \emph{overlapping} plateau models if they belong to the same class. The parameters of the merger are set to cover the union of the regions covered by the two previous plateau models. In this revision, the details on the merge process have been added in Algorithm 1 (p. 6) and Appendix C.
>
> **3-2. What is the computational complexity and/or time for each round? Is there a way to lower the complexity?**
>
> In each round of active learning, let $M$ be the number of plateau models, $V$ be the average length of sub-sequences of predicted probability $f_\theta(x_t)[k]$, and $S$ be the number of training steps for evaluating Eq. (4). Then, considering constant computational complexity for calculating the loss and gradient at each timestamp in the sub-sequences, we derive the computational complexity of plateau model fitting per round to be $\mathcal{O}(MVS)$. Here, $M$ can be reduced by merging two overlapping plateau models with the same class. This complexity of fitting the plateau models is negligible compared with the complexity of training the classifier. For instance, in an experiment done using Intel Xeon Gold 6226R and Nvidia RTX3080 against the 50salads dataset, fitting the plateau models took only about 1 to 2 minutes, whereas training the classifier took about half an hour per active learning round.
> In this revision, we have added the computational complexity analysis in Appendix A.
>
> **4. The same settings on $b$ and $w_0$ should be used across all datasets, and it would be better to state some rules for selecting them.**
>
> In this revision, we have changed the value of $b$ for the mHealth dataset to $200$ in order to make it consistent for all the datasets, and obtained similar results that TCLP again shows the best performance; the new results have been incorporated into Table 2, Table 3, and Figure 4.
> On the other hand, we have kept the value of $w_0$ for the GTEA dataset to be the same as before (i.e., $5$) because the value should better be tailored to the dataset for accurate label propagation when the plateau models are initialized. Our experience indicates that any value of $w_0$ smaller than 20% of the mean segment length is adequate enough for TCLP to work well. We briefly mentioned this rule in the revision (for hyperparameter selection in p. 7).

---

> > ### Comment · Reviewer_s4ST · 2021-11-21
> > **Suggested proof of validity of probability is still weak**
> >
> > Dear Authors, thanks for your responses to my comments.
> >
> > On point 2 (above) related to proving that Eqn 5 is a true probability, the suggested proof assumes that L -> inf. This implies that we must assume that the true length of a segment always approaches infinity (as per definition of 'L'). But this assumption would make the algorithm theoretically weak and impose severe constraints such as not being able to segment shorter or even moderate duration events. I am not sure the limit is justified. I would expect the probabilities to add up to 1 irrespective of L -> inf.

---

> > > ### Author Response · Authors · 2021-11-22
> > > **Additional Response to Reviewer s4ST**
> > >
> > > Thank you very much for your careful review. We would like to justify our formulation as follows.
> > >
> > > First of all, we would like to emphasize that Eq. (5) is not related to TCLP but to a threshold-based approach, which was introduced as a baseline for theoretical comparison.
> > >
> > > Using Eq. (5), we intend to prove that the threshold-based approach is likely to fail to estimate the true segment of a length $L$, especially when $L$ is sufficiently large. Here, the probability in Eq. (5) needs to be considered for all "**possible**" lengths of the true segment, and thus $L$ can approach $\infty$. But in practice, an unrealistically large length will not likely be observed because of the negligible probability in Eq. (5).
> > >
> > > Overall, we do **not** assume that $L$ should always approach $\infty$, but we just allow $L$ to approach $\infty$. We have modified our original response accordingly (see the above) and hope that our answer will resolve your concerns.

---

> > > > ### Comment · Reviewer_s4ST · 2021-11-23
> > > > **Not quite satisfactory theory**
> > > >
> > > > First, the 'z' in Eqns (5), (6), (7), (8) are dependent on the class k (by definition). Hence, it would be more appropriate to denote them by z_k.
> > > > Second, the definition of 'z' (P(. >= 0.5) ) in (7) and (8) is slightly different from (5) and (6) (P(. >= \delta)).
> > > >
> > > > Now, in the 'comparison and discussion' section, it is pointed out that the threshold method converges toward a particular ratio and is not affected by L. However, note that in this case the ratio is dependent on z_k -- hence the convergence is toward a different length for each class. It is likely that 'L' indirectly acts through z_k and the point of convergence might be fine in reality -- and if so, then does the threshold method still suffer from the shortcoming pointed out? I feel the theory is still not rigorous enough to answer all questions.
> > > >
> > > > Another issue I find is that in Eqn 5 the basis for z^(l-1)(1-z)^2 has not been clarified. I am wondering if the assumption is that the length is expanded in one direction with each time-step and this expansion is considered completely independent of previous expansions. If so, I find this approach to be very sketchy. It should be more accurate to consider the probability conditional of the previous time step.

---

> > > > > ### Author Response · Authors · 2021-11-26
> > > > > **Additional Response to Reviewer s4ST**
> > > > >
> > > > > **1. First, the $z$ in Eqns (5), (6), (7), (8) are dependent on the class $k$ (by definition). Hence, it would be more appropriate to denote them by $z_k$. Second, the definition of $z$ ($P(. >= 0.5)$ ) in (7) and (8) is slightly different from (5) and (6) ($P(. >= \delta)$).**
> > > > >
> > > > > Thank you for the further review on theoretical analysis.
> > > > >
> > > > > We agree. Since $z$ is defined for a given $k$, we will denote $z$ as $z_k$; in addition, we will set $\delta$ to $0.5$ in Eqns. (5) and (6).
> > > > >
> > > > > **2. Now, in the 'comparison and discussion' section, it is pointed out that the threshold method converges toward a particular ratio and is not affected by $L$. However, note that in this case the ratio is dependent on $z_k$ -- hence the convergence is toward a different length for each class. It is likely that '$L$' indirectly acts through $z_k$ and the point of convergence might be fine in reality -- and if so, then does the threshold method still suffer from the shortcoming pointed out? I feel the theory is still not rigorous enough to answer all questions.**
> > > > >
> > > > > Let us add a further clarification. There are multiple segments in a time series, and they may well have different segment lengths regardless of whether their class labels are the same or not. So, while we can say that a segment's length ($E_{f_\theta(x_t)}[t_e-t_s]$ in Eq. (6)) is dependent on the class $k$, we cannot say it is dependent on $z_k$ because (1) $z_k$ ($=\text{Pr}(f_\theta(x_t)[k] \geq \delta)$) is only the probability that the estimated probability the class being $k$ at timestamp $t$ exceeds the threshold $\delta$ and (2) this class probability is assumed to be conditionally independent of the timestamp. Accordingly, we would like to assert that the shortcoming we pointed out is true of the threshold-based approach.
> > > > >
> > > > > **3. Another issue I find is that in Eqn 5 the basis for $z^{(l-1)}(1-z)^{2}$ has not been clarified. I am wondering if the assumption is that the length is expanded in one direction with each time-step and this expansion is considered completely independent of previous expansions. If so, I find this approach to be very sketchy. It should be more accurate to consider the probability conditional of the previous time step.**
> > > > >
> > > > > As we described in Section 3.3 of the revised paper, we assume conditional independence between estimated class probabilities at different timestamps. This assumption is based on a widely-used sequence classification framework (Graves et al., 2006;Chung et al., 2015). Letting $Y= \{ f_\theta(x_1)[k],\ldots,f_\theta(x_L)[k]) \}$ and $X=\{x_1, \ldots, x_L\}$, the output probabilities $f_\theta(x_t)[k]$ $(t = 1, 2, ..., L)$ in $Y$ are conditionally independent of one another when the input $X$ is given according to the assumption. The reasoning behind this assumption is that $f_\theta(x_i)[k]$ is not directly used to derive $f_\theta(x_j)[k]$, where $i \neq j$, and the classifier $f_\theta$ is deterministic. Under the conditional independence assumption, we can derive $z^{l-1}(1-z)^2$ as follows.
> > > > >
> > > > > The threshold-based approach first checks the estimated probabilities at timestamps to the right of the center $t_q$ and extends the segment to the right as long as the condition $f_\theta(x_t)[k]>\delta$ is satisfied at each timestamp $x_t$ (and stops at the first timestamp where the condition is not satisfied). Then, the probability of having consecutive $n$ timestamps satisfying the condition on the right side is $z^{n}(1-z)$. Similarly, the probability in the left direction is $z^{m}(1-z)$ where $m$ is the number of consecutive timestamps that satisfy the condition on the left side. Thus, the total number $l$ of consecutive timestamps satisfying the condition is $l=m+n+1$ (where $1$ is for the center timestamp). By the multiplication law of probability, the probability of consecutive $l$ timestamps satisfying the condition is $z^{m+n}(1-z)^2=z^{l-1}(1-z)^2$.
> > > > >
> > > > > **REFERENCES**
> > > > > 1. Junyoung Chung, Kyle Kastner, Laurent Dinh, Kratarth Goel, Aaron C Courville, and Yoshua Bengio.  A recurrent latent variable model for sequential data.  In Advances in Neural Information Processing Systems 28: Annual Conference on Neural Information Processing Systems (NIPS), pp. 2980--2988, December 2015.
> > > > > 2. Alex Graves, Santiago Fern ́andez, Faustino Gomez, and J ̈urgen Schmidhuber.  Connectionist temporal classification: Labelling unsegmented sequence data with recurrent neural networks. In Proceedings of the 23rd International Conference on Machine Learning (ICML), pp. 369--376, June 2006.

---

### Official Review · Reviewer_hW9X · 2021-11-04

**Correctness:** 4
**Technical Novelty And Significance:** 4
**Empirical Novelty And Significance:** 4
**Recommendation:** 10
**Confidence:** 4

**Main Review:**

The strengths of this paper are in addressing time series active learning in a way that truly respects the temporal coherence of time series and addressing issues with identifying the time series segments. These are presented in a coherent algorithm with pseudocode that is relatively easy to follow. Additionally the empirical results are quite strong. There is also some analysis of the differences in performance between the proposed algorithm and other competitors, as a function of the characteristics of the dataset---this, along with pseudocode, are unfortunately becoming quite rare in machine learning research papers.

I have one issue that would be nice for the authors to address in the paper. Near the end of the experiment section, the authors mention how performance varies as a function of the parameter T in temperature scaling, and show performance increasing as T increases from 1 to 1.5 to 2. Please describe what happens as T is increased further until the point at which performance decreases. It would be good to see how that breaking point varies depending on the dataset, and why.



**Summary Of The Paper:**

This paper develops a framework for time series active learning called Temporal Coherence-based Label Propagation (TCLP), that uses the temporal coherence within time series to choose as few samples as possible for domain expert labeling and assigns those labels to temporally close samples. In addition, unlike the most similar methods in the literature, the proposed method does not require estimates of the approximate locations and true classes of the segments of the time series.

**Summary Of The Review:**

This paper, as far as this reviewer knows, is truly addressing time series active learning for the first time, and does so in a novel and coherent way algorithmically and experimentally.

---

> ### Author Response · Authors · 2021-11-20
> **Response to Reviewer hW9X**
>
>
> We deeply appreciate the reviewer's constructive comments and positive feedback on our manuscript.
>
> **1. Please describe what happens as $T$ is increased further until the point at which performance decreases. It would be good to see how that breaking point varies depending on the dataset.**
>
> As the reviewer mentioned, $T$ is critical for effective label propagation in TCLP. In this revision, we conducted additional experiments for the 50salads and GTEA datasets, with $T$ varying in a larger range from $0.5$ to $2.75$ at the increment of $0.25$. The break point was reached at $T = 2.0$ according to the average of six query selection methods.
>
> Table 1 summarizes the results. Evidently, "too low" or "too high" $T$ results in performance degradation. In the 50salads dataset, when $T > 2$, the label propagation length is suppressed, thereby causing deficiency in labels needed to train a classifier; when $T < 1$, the label propagation length may exceed the true segment length, thereby including wrong labels when training the classifier.
>
> The break point is affected by the average length of segments in each dataset, where it occurs at a higher value of $T$ in a dataset with shorter segments: the average segment length is 34 in the GTEA dataset whereas it is 289 in the 50salads dataset.
>
> In this revision we enriched the content of Table 4 (included below as well) and discussed it in Section 4.4 (p. 9).

---

> > ### Author Response · Authors · 2021-11-20
> > **Additional Response to Reviewer hW9X**
> >
> > | Dataset  | Width Reg. and Temp. Scal. | No, No($T=1$)     | No, Yes($T=2$)     | Yes, $T=0.5$      | Yes, $T=0.75$     | Yes, $T=1$        | Yes, $T=1.5$          | Yes, $T=1.75$         | Yes, $T=2$            | Yes, $T=2.25$         | Yes, $T=2.5$          | Yes, , $T=2.75$       |
> > |----------|----------------------------|-------------------|--------------------|-------------------|-------------------|-------------------|-----------------------|-----------------------|-----------------------|-----------------------|-----------------------|-----------------------|
> > | 50salads |  CONF                      |  0.441$\pm$0.015  |  0.487$\pm$0.030   |  0.465$\pm$0.044  |  0.489$\pm$0.026  |  0.480$\pm$0.035  |  0.519$\pm$0.019      |  0.559$\pm$0.020      |  **0.559$\pm$0.010**  |  0.535$\pm$0.020      |  0.508$\pm$0.023      |  0.460$\pm$0.045      |
> > | 50salads |  ENTROPY                   |  0.431$\pm$0.042  |  0.455$\pm$0.040   |  0.430$\pm$0.044  |  0.410$\pm$0.027  |  0.442$\pm$0.013  |  0.462$\pm$0.029      |  0.452$\pm$0.009      |  **0.496$\pm$0.027**  |  0.479$\pm$0.015      |  0.462$\pm$0.018      |  0.482$\pm$0.041      |
> > | 50salads |  MARG                      |  0.655$\pm$0.033  |  0.671$\pm$0.027   |  0.668$\pm$0.033  |  0.667$\pm$0.016  |  0.691$\pm$0.025  |  0.671$\pm$0.024      |  0.682$\pm$0.018      |  **0.697$\pm$0.020**  |  0.664$\pm$0.028      |  0.658$\pm$0.013      |  0.599$\pm$0.030      |
> > | 50salads |  CS                        |  0.611$\pm$0.028  |  0.618$\pm$0.026   |  0.592$\pm$0.040  |  0.616$\pm$0.027  |  0.624$\pm$0.034  |  0.610$\pm$0.020      |  0.627$\pm$0.029      |  **0.657$\pm$0.024**  |  0.656$\pm$0.018      |  0.633$\pm$0.025      |  0.626$\pm$0.015      |
> > | 50salads |  BADGE                     |  0.595$\pm$0.018  |  0.599$\pm$0.020   |  0.575$\pm$0.021  |  0.594$\pm$0.037  |  0.623$\pm$0.023  |  0.631$\pm$0.017      |  **0.634$\pm$0.014**  |  0.600$\pm$0.025      |  0.575$\pm$0.018      |  0.546$\pm$0.028      |  0.566$\pm$0.010      |
> > | 50salads |  UTILITY                   |  0.671$\pm$0.017  |  0.670$\pm$0.016   |  0.662$\pm$0.024  |  0.644$\pm$0.036  |  0.662$\pm$0.022  |  0.652$\pm$0.037      |  0.651$\pm$0.024      |  **0.672$\pm$0.018**  |  0.661$\pm$0.024      |  0.661$\pm$0.025      |  0.667$\pm$0.028      |
> > | 50salads |  AVERAGE                   |  0.567$\pm$0.039  |  0.583$\pm$0.034   |  0.565$\pm$0.037  |  0.570$\pm$0.037  |  0.587$\pm$0.038  |  0.591$\pm$0.031      |  0.601$\pm$0.031      |  **0.614$\pm$0.028**  |  0.595$\pm$0.029      |  0.578$\pm$0.032      |  0.566$\pm$0.030      |
> > | GTEA     |  CONF                      |  0.539$\pm$0.072  |  0.575$\pm$0.052   |  0.614$\pm$0.025  |  0.587$\pm$0.020  |  0.603$\pm$0.023  |  0.607$\pm$0.010      |  0.575$\pm$0.011      |  **0.654$\pm$0.011**  |  0.553$\pm$0.082      |  0.638$\pm$0.027      |  0.477$\pm$0.087      |
> > | GTEA     |  ENTROPY                   |  0.553$\pm$0.016  |  0.578$\pm$0.020   |  0.606$\pm$0.016  |  0.574$\pm$0.018  |  0.596$\pm$0.018  |  **0.614$\pm$0.019**  |  0.592$\pm$0.022      |  0.590$\pm$0.021      |  0.582$\pm$0.026      |  0.551$\pm$0.022      |  0.593$\pm$0.010      |
> > | GTEA     |  MARG                      |  0.605$\pm$0.014  |  0.646$\pm$0.023   |  0.609$\pm$0.015  |  0.636$\pm$0.014  |  0.632$\pm$0.010  |  0.636$\pm$0.014      |  0.672$\pm$0.009      |  0.659$\pm$0.015      |  **0.682$\pm$0.021**  |  0.669$\pm$0.009      |  0.657$\pm$0.032      |
> > | GTEA     |  CS                        |  0.596$\pm$0.012  |  0.606$\pm$0.013   |  0.601$\pm$0.021  |  0.586$\pm$0.012  |  0.484$\pm$0.112  |  0.598$\pm$0.021      |  0.597$\pm$0.019      |  0.630$\pm$0.007      |  0.616$\pm$0.020      |  0.629$\pm$0.015      |  **0.673$\pm$0.011**  |
> > | GTEA     |  BADGE                     |  0.645$\pm$0.015  |  0.644$\pm$0.011   |  0.616$\pm$0.012  |  0.635$\pm$0.018  |  0.620$\pm$0.010  |  0.641$\pm$0.016      |  0.658$\pm$0.015      |  0.663$\pm$0.015      |  0.676$\pm$0.019      |  **0.687$\pm$0.017**  |  0.677$\pm$0.007      |
> > | GTEA     |  UTILITY                   |  0.589$\pm$0.043  |  0.608$\pm$0.031   |  0.605$\pm$0.011  |  0.529$\pm$0.081  |  0.616$\pm$0.016  |  0.595$\pm$0.017      |  0.627$\pm$0.017      |  0.644$\pm$0.019      |  0.638$\pm$0.014      |  0.659$\pm$0.015      |  **0.667$\pm$0.013**  |
> > | GTEA     |  AVERAGE                   |  0.588$\pm$0.014  |  0.609$\pm$0.012   |  0.608$\pm$0.002  |  0.591$\pm$0.015  |  0.592$\pm$0.020  |  0.615$\pm$0.007      |  0.620$\pm$0.014      |  **0.640$\pm$0.010**  |  0.624$\pm$0.019      |  0.639$\pm$0.018      |  0.624$\pm$0.029      |

---

### Author Response · Authors · 2021-11-20
**Response to All Reviewers**

Dear Reviewers, thank you so much for your effort to review the paper. We have prepared our rebuttal (in sections below) and revised the manuscript carefully according to the comments. The revised parts are highlighted in color and labeled with the reviewer ID (R1: hW9X, R2: s4ST, R3: pVjg, and R4: kJBx).

---

### Decision · Program_Chairs · 2022-01-20

**Decision:**

Accept (Poster)

**Comment:**

The authors design a framework for active learning on time-series data. The framework, called Temporal Coherence-based Label Propagation (TCLP) leverages temporal coherence to propagate expert labels to nearby points by a plateau model. In addition to describing the framework clearly with simple pseudocode, several experiments are carried out with careful analysis to validate the effectiveness of the framework.

The reviewers are mostly positive on the simple algorithm with strong empirical performance as well as the solid analysis of experimental results. They are also satisfied with most of the rebuttal feedback from the authors. Somehow there are joint concerns on the weaker theoretical results, especially in terms of their correctness. In particular, the unrealistic assumptions and over-simplification make it hard to connect the theoretical results with the actual algorithm. Several reviewers suggest the authors to move theoretical analysis section to a supplementary section as a hypothesis, and the authors are also encouraged to clearly discuss what the theory can and cannot cover.